# On the Impact of Maximum Speed on the Power Density of Electromechanical Powertrains

**Daniel Schweigert** [1,*], **Martin Enno Gerlach** [2], **Alexander Hoffmann** [2], **Bernd Morhard** [1], **Alexander Tripps** [1], **Thomas Lohner** [1], **Michael Otto** [1], **Bernd Ponick** [2] **and Karsten Stahl** [1]

[1] Gear Research Centre (FZG), Technical University of Munich, Boltzmannstraße 15, 85748 Garching near Munich, Germany; morhard@fzg.mw.tum.de (B.M.); alexander.tripps@tum.de (A.T.); lohner@fzg.mw.tum.de (T.L.); otto@fzg.mw.tum.de (M.O.); stahl@fzg.mw.tum.de (K.S.)

[2] Institute for Drive Systems and Power Electronics, Leibniz University Hannover, 30167 Hannover, Germany; martin.gerlach@ial.uni-hannover.de (M.E.G.); alexander.hoffmann@ial.uni-hannover.de (A.H.); ponick@ial.uni-hannover.de (B.P.)

* Correspondence: schweigert@fzg.mw.tum.de; Tel.: +49-89-289-15774

**Abstract:** In order to achieve the European Commission's ambitious climate targets by 2030, BEVs (Battery Electric Vehicles) manufacturers are faced with the challenge of producing more efficient and ecological products. The electromechanical powertrain plays a key role in the efficiency of BEVs, which is why the design parameters in the development phase of electromechanical powertrains must be chosen carefully. One of the central design parameters is the maximum speed of the electric machines and the gear ratio of the connected transmissions. Due to the relationship between speed and torque, it is possible to design more compact and lighter electric machines by increasing the speed at constant power. However, with higher speed of the electric machines, a higher gear ratio is required, which results in a larger and heavier transmission. This study therefore examines the influence of maximum speed on the power density of electromechanical powertrains. Electric machines and transmissions with different maximum speeds are designed with the state-of-the-art for a selected reference vehicle. The designs are then examined with regard to the power density of the overall powertrain system. Compared to the reference vehicle, the results of the study show a considerable potential for increasing the power density of electromechanical powertrains by increasing the maximum speed of the electric machines.

**Keywords:** E-Mobility; powertrain design; high-speed; electric machine design; transmission design; gearbox

## 1. Introduction

In 2011, the EU Commission for Energy, Climate change and Environment published ambitious plans for the future climate and energy policy framework. By 2030, greenhouse gas emissions have to be lowered by at least 40% in comparison to 1990. Furthermore, the energy efficiency of products should, at the same time, increase by at least 32.5%. Limit values for $CO_2$ emissions of newly registered cars of each manufacturer are also part of this initiative. The average emissions of a manufacturer's fleet in 2021 must, for example, be lower than the limit of 95 grams of $CO_2$ per 100 km [1]. If manufacturers exceed the limit values for $CO_2$ emissions, they are expected to pay substantial fines. In order to achieve the ambitious goals, set for the protection of the environment and to avoid heavy fines, the mobility behavior has to change significantly.

The electrification of the automotive powertrain is expected to play an important role in this change. A combination of battery electric vehicles (BEVs) and electricity from renewable energy

sources can enable a $CO_2$ neutral and largely pollutant-free usage phase and thus make a significant contribution to reducing greenhouse gas emissions in the future. Furthermore, the tank-to-wheel efficiency of electromechanical powertrains in BEVs, which is significantly higher than that of internal combustion engines (ICE), is intended to make a contribution to the increase of efficiency of mobility [1].

The electromechanical powertrain of a BEV typically consists of power electronics, a drive motor or electric machine and a transmission. While the power electronics are responsible for the conversion of electrical power to control the electric motor, the transmission's gear ratio is used to adapt the electric machine's speed and torque to the required values at the drive axle. An important parameter in the design process of an electromechanical powertrain is the maximum speed of the electric machines in combination with the gear ratio of the transmission to reach the desired maximum speed of the vehicle. A higher speed of the electric machine leads to a lower required torque at constant power, which means that more compact and lighter electric machines can be designed. However, a transmission with an increased input speed needs a higher gear ratio to reach the same output speed and must therefore be designed larger and heavier. Nevertheless, there is a trend towards increasing speeds of electric machines in a BEV. The positive influence of increased speeds of the electric machine on the power density of the whole powertrain has been proven in various studies [2,3]. This positive influence may play an important role in the achievement of ultimate efficiency and climate goals by 2030.

This study presents a detailed analysis of the influence of maximum speed of the electric machine on the power density of an electromechanical powertrain. For this purpose, conceptual designs of electric machines with various maximum speeds and transmissions with suitable overall gear ratios have been developed. The considered speeds are set from $n = 12,000$ min$^{-1}$ to $n = 50,000$ min$^{-1}$. While the speed increases and thus the torque of the electric machines decreases, the gear ratio has to increase to remain a constant output torque and vehicle speed. These boundary conditions are based on the parameters of the BMW i3 (120Ah) as reference vehicle. Table 1 shows the parameters of the electric machine and transmission relevant for this study. The voltage level of the battery is increased to satisfy the demands of the high-speed electric machine.

**Table 1.** Relevant parameters and view of the BMW i3 reference vehicle (120Ah). Data from [4].

| BMW i3 (120 Ah) | | |
|---|---|---|
| Empty weight | 1245 kg | |
| Top speed | 150 km/h | |
| Rated power | 75 kW | |
| Peak power | 135 kW | |
| Battery voltage | 660 V | |
| Rated torque (EM) | 150 Nm | |
| Peak torque (EM) | 250 Nm | |
| Gear ratio | 9.665 | |
| Max. speed (EM) | 11,400 min$^{-1}$ | |
| Max. speed (drive axle) | 1179.5 min$^{-1}$ | |
| Rated torque (drive axle) | 1449.75 Nm | |
| Max. torque (drive axle) | 2416.25 Nm | |

The electric machine of the reference vehicle is designed as a permanent magnet synchronous machine (PMSM) with a rated power of 75 kW and a peak power of 135 kW. Furthermore, the electric machine has a rated torque of 150 Nm and a peak torque of 250 Nm and reaches a maximum speed of 11,400 1/min. The gear ratio determines the corresponding parameters on the drive axle. All of the designed electric machines and transmissions for every considered maximum speed and overall gear ratio result in a large number of electromechanical powertrains. All of these designs represent potential drivetrains for the reference vehicle and will be analyzed with regard to mass, volume, volumetric and gravimetric power density. This is intended to make a statement regarding the potential of increase of the maximum speed in electromechanically powertrains to increase power density.

After applying the state of the art to design parameters of high-speed electric machines and transmissions in Sections 2–4 present the conceptual design process of the electric machines and transmissions used in this study. The results of the study will be shown in Section 5, followed by a discussion and an outlook of the results in Section 6.

## 2. State of the Art Considerations for Higher Shaft Speeds

The following section will cover the key aspects of current developments in the field of electrical machines and transmissions with regard to high speeds. After discussing the current technological driver in this field, this section will close with a summary of current and future generation powertrains.

### 2.1. Electric Machines

In the context of electrical machines, the term high-speed refers to the surface speed of the rotor $v_s$ and not to the synchronous shaft speed

$$n_0 = \frac{f_1}{p},\tag{1}$$

where $f_1$ is the fundamental electric stator frequency and $p$ is the number of pole pairs. It is therefore important to understand the relation

$$v_{\mathrm{rot}} = 2\pi r_{o,2} n,\tag{2}$$

where $r_{o,2}$ is the outer radius of the rotor, because the surface speed $v_{\mathrm{rot}}$ is in fact the variable on which the mechanical boundaries depend in the first place. The variable on which the electrical boundaries depend in the first place is the fundamental electric stator frequency $f_1$. With the importance of these two quantities in mind, subjects such as rotor topologies, winding technologies and materials will be discussed in the context of PMSMs [5].

#### 2.1.1. Rotor Topologies

Rotors for PMSMs are subdivided into two categories, surface type rotors and interior type rotors. The definition is based on the location of the permanent magnet relative to the outer contour of the rotor iron, as shown in Figure 1. Depending on the desired maximum surface speed, surface PM rotors are preferable to interior PM rotors [6].

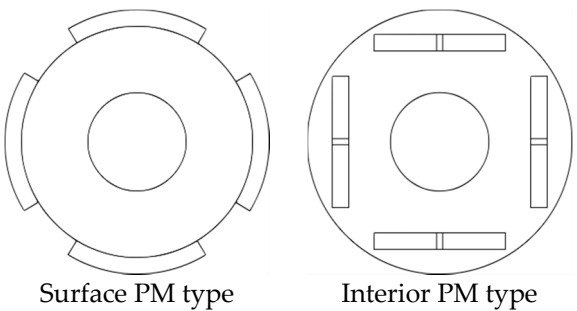

Surface PM type    Interior PM type

**Figure 1.** Two different rotor topologies for PMSMs.

The mechanical stresses introduced on the rotor iron due to rotation are proportional to the square of the rotational speed and the square of the outer radius $r_{o,2}$. Due to this strongly non-linear relationship, the mechanical stress on the rotor iron requires special consideration. The introduction of so-called pockets for the permanent magnets has the consequence of weakening the material, which further increases the mechanical stresses [7]. For fast rotating electric machines, it is common to reinforce the rotor with a so-called bandage or sleeve. Such reinforcement is applied on the outer radius and prevents the expansion of the rotating rotor. The insertion of a bandage thus has the consequence of increasing the magnetically effective air gap. A larger magnetic air gap acts like an

increased magnetic resistance and reduces the magnetic flux, which has an influence on a number of important parameters of the electric machine. The use of a bandage must therefore be weighed against other possibilities and the influence on all requirements must be checked. This includes the selection of the bandage itself. Different materials will be discussed in a later section. Decisions are usually made by evaluating results from finite element simulations [8].

### 2.1.2. Rotor Materials

To go into more detail, an overview of materials is presented for the three basic components (besides the shaft) of a PMSM rotor, electrical steel sheets, permanent magnets and bandage material. An example for a high-speed PMSM rotor is shown in Figure 2.

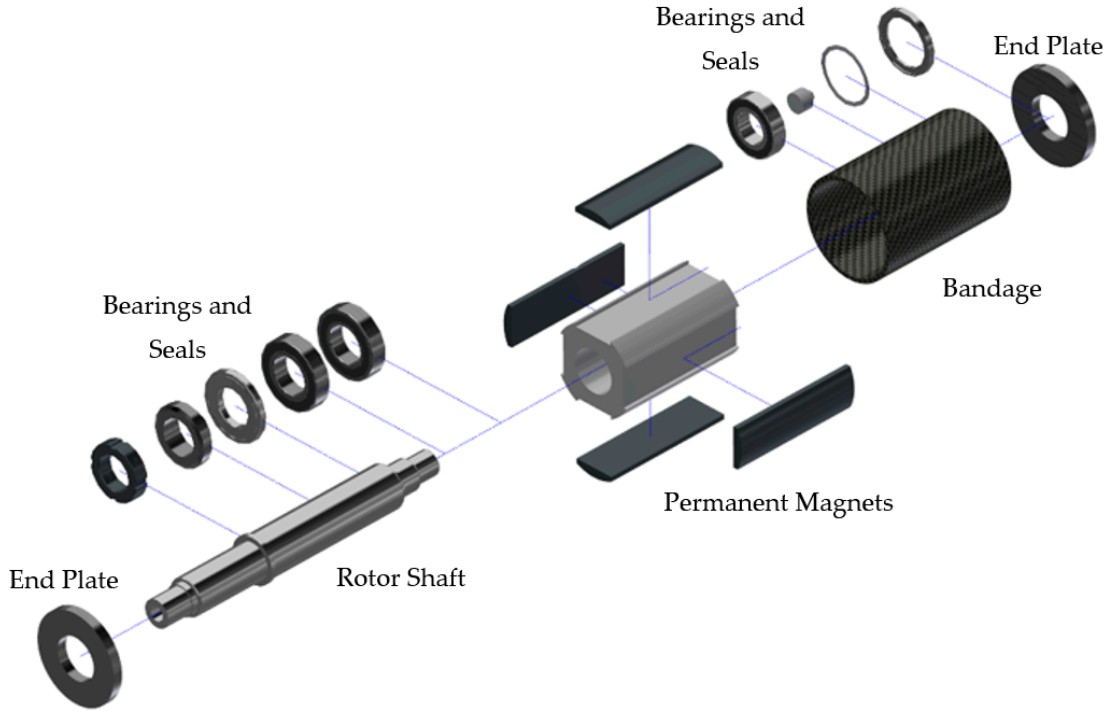

**Figure 2.** Exploded view of an PMSM rotor assembly.

### 2.1.3. Electrical Steel Sheets

While industrially used induction machines usually have a laminated rotor, PMSMs do not require a laminated rotor. However, the use of electric sheet metal has a positive effect on the losses of the rotor due to eddy currents and therefor on the rotor temperature. Electrical steel is the most common material seen in electrical machines; with increasing demands on efficiency and power density, electrical steel with cobalt is gaining interest. However, the cost of cobalt electrical steel is considerably higher. High strength steel is particularly interesting for the rotors electrical machines with high speeds [9]. The yield strength of such sheets is higher than that of electrical steel sheets but returns less favorable magnetic properties. Recent developments in electrical steel have resulted in high-strength electrical steel, which aims to close the gap between conventional electrical steel and structural optimized steel [10].

### 2.1.4. Permanent Magnets

In addition to the different materials for permanent magnets (AlNiCo, NdFeB and SmCo), a variety of different segmentation strategies exist. Segmentation of permanent magnets is performed in order to reduce eddy currents inside of the permanent magnets and to avoid overheating of the rotor assembly [11]. If the rotor in a PMSM reaches unconsidered temperatures, a loss of torque at the shaft

and permanent damage due to demagnetization can occur. Currently, permanent magnet segmentation can be performed with a layer thickness of as low as 500 μm in thickness and a <20 μm insulation between the layers [12]. In general, segmentation can be in axial or in tangential direction of the rotor. The selection of suitable permanent magnet shapes and the placement within the rotor are most often aided by magnetostatic simulations using the finite-element-method [13].

### 2.1.5. Sleeve and Bandage Materials

Materials for sleeves and bandages can be differentiated into steel-based sleeves and composite-based sleeves. Materials for steel sleeves are high strength steels with alloying elements such as chromium, molybdenum and nickel. For steel-based sleeves, the highest tensile strength is given at 700 N/mm$^2$ for 2.461 NiMo16Cr16Ti; in comparison, the well-known 1.7225 42CrMo4 achieves a tensile strength of 550 N/mm$^2$ according to the manufacturer's data [14,15]. Both have a good strength to weight ratio and are considerably stronger and harder than standard steels. A side-by-side comparison of the values for tensile strength, density and electrical resistivity shows that the composite-based sleeves are more suitable for high-speed electrical machines than sleeves made of steel. It should be said, that the properties of the fiber and the finished composite sleeve depend heavily on the fabrication process. Therefore, the given data in Table 2 is for bare fiber and is presented to provide a starting point for further analysis.

**Table 2.** Bare fibers for composite-based sleeve materials. Data from [16].

| Material | Glass Fiber | Aramid Fiber | Carbon Fiber |
|---|---|---|---|
| Tensile strength in N/mm$^2$ | 3400 | 2880 | 3950 |
| Density in kg/m$^3$ | 2600 | 1450 | 1760 |
| Electrical resistivity in Ωmm$^2$/m at 20 °C | $1 \times 10^{20}$ | $1 \times 10^{20}$ | $1.6 \times 10^7$ |
| Typical operating temperature in °C | below 100 | below 200 | below 1000 |
| Thermal expansion coefficient in $10^{-6}$ 1/K | 5 | −3.5 | −0.1 |

### 2.1.6. Winding Technologies

In the field of traction motors, two different types of distributed windings can be observed for electrical machines. The random winding, consisting of randomly distributed round shaped conductors, and the form wound winding, which is composed of so-called hairpins. Hairpins in the context of windings are solid copper conductors with rectangular shape, are formed into the shape of a conventional hairpin by bending. By connecting several hairpins trough welding, the actual winding is created. Hairpin windings are favored by car manufacturers because the manufacturing process can be fully automated [17]. But this does not mean that all traction motors for electric cars are equipped with hairpin windings. Both winding technologies are still on the market. In the context of high-speed electric machines, hairpin windings have concrete disadvantages. The cross-sectional area of a hairpin must not fall below a certain value, otherwise processing becomes more difficult. Since the winding resistance has a great influence on efficiency, and since the phase current in high-speed electrical machines can reach higher frequencies than in conventional designs, the frequency dependence of the winding resistance has to be considered for the selection of the winding technology. Finally, Figure 3 shows an example of the phase resistance for the two mentioned winding technologies as a function of frequency. The displayed data is acquired via direct measurements at electrical machines. It can be seen that the resistance of the hairpin wound winding is already greater by a factor of 10 at a frequency of 1000 Hz due to current displacement effects. This factor has a direct linear effect on the power loss in the winding [18].

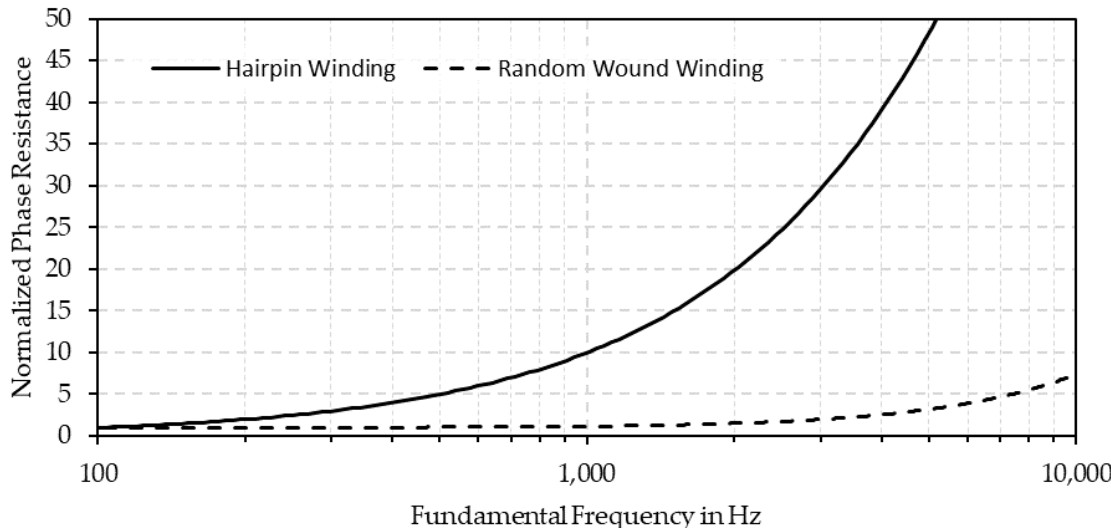

**Figure 3.** Comparison of the phase resistance of a hairpin winding with a conductor diameter of 0.5 mm and a random wound winding with a conductor width of 3.2 mm and a height of 1.6 mm (Phase resistance is related to DC resistance).

### 2.2. Technologies for High-Speed Transmissions

High-speed electromechanical powertrains as scope of recent research history [19–24] promise an increased power density of the whole powertrain. Indeed, the required torque and speed at the wheels cause the transmission to be designed with a higher total gear ratio that on the first glance causes increased weight, installation space and cost of the transmission itself. Still, the conglomerate of high-speed electrical machines and high-speed transmissions is able to demonstrate its benefits. To meet the high-level requirements of BEVs concerning range, comfort and operability new technologies have to support and enable transmissions to operate within this high-speed approach.

### 2.2.1. Improved Efficiency for Maximized Range

As the available range of BEVs is reduced by the power losses of the whole powertrain, the efficiency of the powertrain in general and of the transmission in particular is of prime importance. The power loss of a transmission and hence the efficiency is determined by gear power losses, bearing power losses, sealing power losses and other power losses, e.g., caused by oil pumps. Both, gear power losses and bearing power losses are separated into load-dependent and no-load power losses. In order to reduce the power losses of the transmission, and more precisely the load-dependent gear power losses, low-loss gear geometries and water-containing gear fluids with coefficients of friction smaller than 0.01 [25] (which is referred to as superlubricity [26]) can be used in electromechanical powertrains. As both technologies reduce the frictional losses, they can also be seen as a compensation for the rising sliding speeds with higher input speeds, usually causing a higher risk of scuffing. The low-loss gear geometry concentrates the path of contact to a minimum around the pitch point. As sliding speed rises with increasing distance to the pitch point, high sliding speeds are avoided this way, resulting in reduced load-dependent gear power losses [27]. Hinterstoisser et al. [28] outline in experiments the significant impact of the low-loss gear geometry on the efficiency. With an extreme low-loss gear design, the power losses can be reduced by about 79% compared to a conventional gear design. In order to reduce the mean gear coefficient of friction, water-containing gear fluids promise a significant improvement of the efficiency. Yilmaz et al. [29] demonstrate on the FZG gear efficiency test rig the reduction of the mean gear coefficient of friction and hence of the load-dependent gear power losses by water-containing fluids (cf. Figure 4). Using those fluids, the mean gear coefficient of friction is reduced by up to 82%.

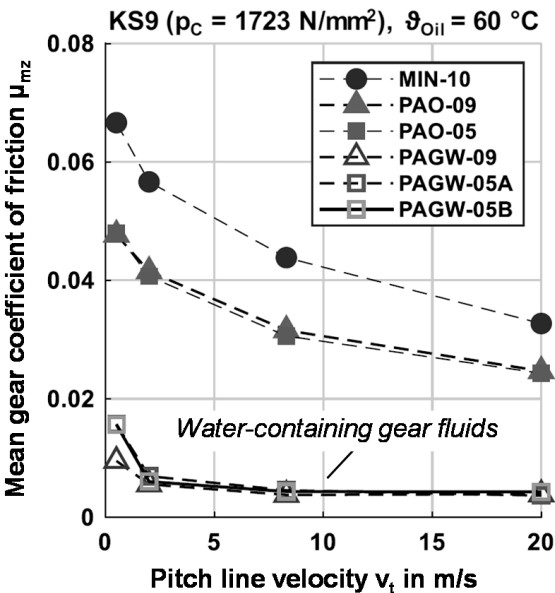

**Figure 4.** Mean gear coefficient of friction $\mu_{mz}$ determined on the FZG gear efficiency test rig at a load $p_C$ of 1723 N/mm$^2$ for a mineral oil (MIN), polyalphaolefine oils (PAO) and water-containing polyalkylenglycols (PAGW) according to Yilmaz et al. [25].

Concerning bearings, investigations with water-containing gear fluids on the FZG bearing power loss test rig show reduced no-load bearing losses and increased load-dependent bearing losses with higher rotational speeds of roller bearings [30] which additionally supports the approach of high speeds and low loads on the input of high-speed transmissions. The authors mention that hybrid bearings with Si$_3$N$_4$ ceramic cylindrical rollers, cronidur races, and polyether ether ketone (PEEK) cages were used to avoid incompatibilities with the investigated water-containing gear fluids [30]. As the presence of hydrogen is suspected to cause white edge cracks (WECs), or at least to support its formation, causing premature failure of the bearings [31–33], WECs have to be considered when using water-containing fluids. As the water content strongly improves its caloric properties, water-containing gear fluids are possible to use as coolant. By this, the whole powertrain including power electronics, electrical machines and the transmission can be cooled and lubricated by one circuit. This holistic thermal management promises further improvement of efficiency of the powertrain as e.g., injection lubrication in the transmission can be used without additional oil pumps.

### 2.2.2. Improved Acoustics for Increased Comfort

As the internal combustion engine is cut in BEVs, its masking sound is not present anymore. Consequently, the customer faces noise of the transmission in BEVs which can be felt as uncomfortable, particularly because of its tonal character. Furthermore, in comparison to conventional powertrains in ICE, the drive speeds are already higher in BEV series solutions, which leads to new spectral compositions of the noise emissions. In particular, the high-frequency components of the spectrum can be perceived as disturbing by the human ear. The increase in speed in comparison to BEV series solutions, which is carried out in the context of this study to achieve higher power densities, further aggravates this problem [34].

In addition, increasing speeds make it more difficult to operate the transmission subcritically over the entire operating range. The subcritical operating range is the range with a mesh frequency f lower than the torsional natural frequency $f_n$ of the meshing. Especially in the critical operating range with reference speeds ($f / f_n$) from 0.85 to 1.15, significant additional dynamic forces and thus vibrations and noise emissions occur in the meshing, as stated in Figure 5. Therefore, the critical operating range

is typically avoided in ICE by operating the transmission in the subcritical range with low dynamic forces [31,35,36].

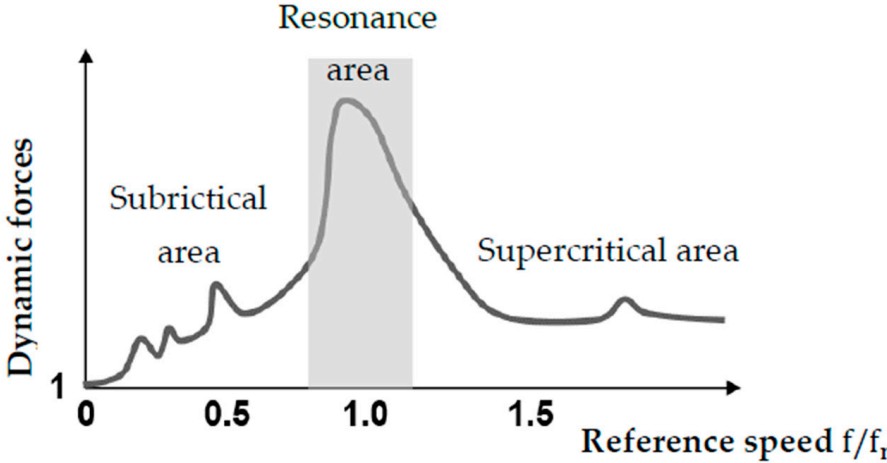

**Figure 5.** Dynamic factor as a function of resonance ratio.

In case of the first stage of the high speed transmissions of this study, subcritical operating will not be possible over the entire operating range with increasing speed of the electric machines. Hence, special precautions are needed to optimize the acoustical behavior of high-speed transmissions. The basis for this optimization is a precise knowledge of acoustically critical operating ranges of all meshes. For powertrains with double-e-architectures, this knowledge can be used to split the required power and torque between two non-identical power-paths and therefore avoid those areas. If there is only one power path available, resonance areas should be positioned in an area which can be passed [35].

### 2.2.3. Operability of High-Speed Transmission

The high circumferential speeds caused by the high rotational speeds on the rotor and input shaft cause stricter requirements the bearings and sealings must fulfill in order to ensure the reliability needed.

That is why the bearings of the input shaft and the rotor of high-speed electrical machines need to be designed for high-speeds. The limiting speed of bearings is given by the speed factor ndm consisting of the maximum speed n [$min^{-1}$] and the mean diameter of the bearing dm [mm]. In the work of Deiml et al. [37] it is mentioned that common SKF bearings are designed with a maximum speed factor of $0.7 \times 10^6$ mm/min, but for their high-speed approach sealed bearings with a speed factor of $1.6 \times 10^6$ were designed. Usually, spindle bearings (super precision angular ball bearings) according to DIN 628-1 [38] are used in machine tools, reaching highest rotational speeds. With oil lubrication, the bearing manufacturer SCHAEFFLER mentions that spindle bearings are suitable for a speed factor of up to over $3.0 \times 10^6$ mm/min [39]. The bearing manufacturer SKF outlines, that hybrid bearings with ceramic balls are superior to all steel bearings in terms of operating speed, as ceramic balls are lighter which causes reduced centrifugal forces leading to reduced losses and heat development [40].

To ensure the operability of the transmission fluid leakage and dirt entry via the input and output shaft have to be avoided by appropriate sealing. Rotary shaft lip seals according to ISO 6194-1 [41], as a part of contact seals and commonly used in automotive sector, operate at circumferential speeds beneath 40 m/s as the frictional losses cause unnecessary high temperatures harming those seals at higher speeds. Contactless seals like labyrinth seals overcome the limiting circumferential speeds as only negligible frictional losses are acting, enabling the operation at highest circumferential speeds [42]. In terms of sealing itself, contactless seals, being more precisely labyrinth seals, are disadvantageous as they are not able to completely seal the transmission, causing leakage [43]. In order to overcome these disadvantages, new sealing technologies have to be used within the high-speed approach.

One promising sealing technology is the gas-lubricated mechanical face seal applied on transmissions. The working principle is that the primer ring and mating ring move apart at rotation of the shaft as a consequence of gas flow (usually air) caused by the aerodynamically optimized surface structure of the mating ring [44]. These seals are characterized by reduced frictional losses, enabling the operation at highest rotational speeds.

### 2.2.4. Research on the High-Speed Transmission of Speed4E

In the joint research project Speed4E [45] a hyper-high-speed powertrain for electric vehicles is developed, designed, and investigated. The goals of Speed4E include the development of an innovative powertrain for BEVs with rotational speeds of the electrical machines of up to 50,000 min$^{-1}$, the integration into a test vehicle, and a holistic thermal management based on a water-containing gear fluid also used for lubrication of the transmission. Different aspects are considered, such as efficiency optimization, mass and cost reduction as well as the power density increase by the high input speeds. In the transmission of the research project Speed4E among other things, the mentioned low-loss gear geometry, NVH-optimized gears, water-containing gear fluids, holistic thermal management, high-speed spindle and hybrid bearings as well as contactless, lifting seals are investigated with respect to input speeds.

### 2.3. Series and Future Axle Drive Systems

A survey on current single axle drive systems for personal BEV shows, that the peak power is around 150 kW and the peak speed is below $n = 20,000$ min$^{-1}$. Depending on which components are included in the axle drive system the weight is at around 80 kg. All shown drive systems in Table 3 aim for a high level of integration and combine the main major parts of a drive system: the electric machine, the inverter of power electronics and the transmission.

**Table 3.** Overview of current axle drive systems from automotive suppliers.

| | Continental (3rd gen.) [46] | Bosch eAxle [47] | Nidec Corporation eAxle [48] | BorgWarner iDM eAxle [49] |
|---|---|---|---|---|
| **Primary use** | Personal BEV | Personal BEV | Personal BEV | Personal BEV |
| **Peak power** | 150 kW | 300 kW | 150 kW | 125 kW |
| **Peak speed** | n/a | 16,000 min$^{-1}$ | 15,000 min$^{-1}$ | 10,600 min$^{-1}$ |
| **Date and status** | In production since 2019 | In production since 2019 | In production since 2018 | In production since 2019 |
| **Package** | PMSM, Inverter and single speed transmission | Electric machine, transmission and Inverter | Electric machine, Inverter and single speed (10.4:1) transmission | Electric motor. single speed transmission (9.4:1) |
| **Weight** | <80 kg | 90 kg | 83 kg | n/a |

## 3. Conceptual Design of High-Speed Electric Machines

In this section, the design process is introduced for electric machines. Four machines with a rated power of $P_N = 75$ kW and different maximum speeds are designed, as it can be seen in Table 4. The PMSM-B1 and PMSM-B2 machines have interior bar magnets and the PMSM-S1 and PMSM-S2 machines have surface mounted magnets with a bandage. The field weakening behavior of machines with interior magnets is better than with surface mounted magnets. Thus, the rated speed of the PMSM-B1 and PMSM-B2 machines is chosen smaller in relation to the maximum speed, than for the PMSM-S1 and PMSM-S2 machines. The geometry, mass, volume, volumetric and gravimetric power density are determined and presented later.

**Table 4.** Speeds and power requirements for the machine designs.

|  | PMSM-B1 | PMSM-B2 | PMSMS-S1 | PMSM-S2 |
|---|---|---|---|---|
| **Rated power $P_N$** |  | 75 kW |  |  |
| **Maximum power $P_{max}$** |  | 135 kW |  |  |
| **Rated speed $n_N$** | 4000 min$^{-1}$ | 6500 min$^{-1}$ | 15,000 min$^{-1}$ | 25,000 min$^{-1}$ |
| **Maximum speed $n_{max}$** | 12,000 min$^{-1}$ | 20,000 min$^{-1}$ | 30,000 min$^{-1}$ | 50,000 min$^{-1}$ |

### 3.1. Design Process of Active Parts

To design an electric machine, different fundamental equations can be used to estimate the size and the geometry of the machine. Therefore, input parameters such as power, speed and voltage have to be provided and assumptions regarding certain machine parameters and boundary conditions have to be specified to start the design process.

Each machine is designed for one specific operating point in the torque speed diagram, i.e., the rated operation, as can be seen in Figure 6. In this case, this point is set to be the point of rated power $P_N$ and rated speed $n_N$ at which the machine's supply voltage $U_1$ reaches the maximum output voltage of the inverter $U_{max}$ and the maximum continuous torque $M_{cont}$ is provided. The machine's induced voltage

$$U_1 \propto w, \hat{\phi}, n, p, \tag{3}$$

which almost matches the supply voltage, is proportional to the magnetic flux $\hat{\phi}$, the speed $n$, the number of pole pairs $p$ and the number of series turns per phase $w$. The torque is proportional to the torque-forming part $I_q$ of the current $I_1$. Using the dq-plane, the current $I_1$ can be displayed as the q-axis current $I_q$ and the magnetizing d-axis current $I_d$ [50]. Since the induced voltage reaches the voltage limit $U_{max}$, for speeds above the rated speed, the current $I_d$ is increased to provide field weakening and decrease the magnetic flux in the machine. Dividing the current $I_1$ by the number of parallel branches per phase $a$, the number of conductors per slot $z$ and the conductors cross sectional area of copper $A_{co}$, the current density

$$S_1 = \frac{I_1}{A_{co} \cdot a} \tag{4}$$

in the slot can be calculated [18]. Depending on the cooling system, the current density $S_{1,cont}$ at which the machine can be operated continuously and the maximum current density $S_{1,max}$ is given. The machine geometry is therefore mainly determined by the magnetic flux $\hat{\phi}$, the current density $S_{1,cont}$, and the maximum voltage $U_{max}$.

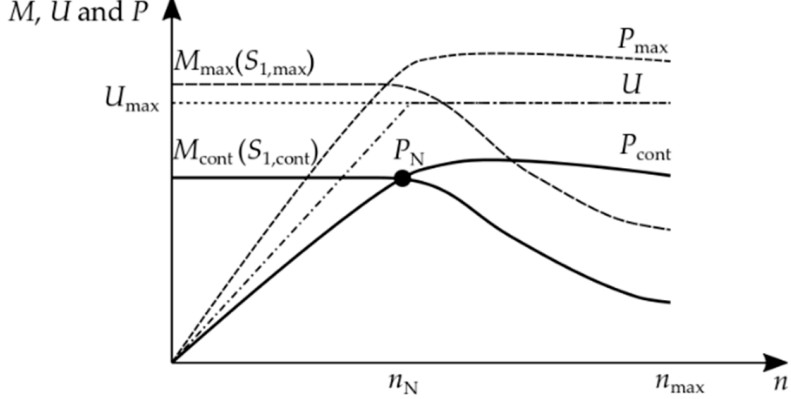

**Figure 6.** Schematic torque speed diagram.

The design process can be seen in Figure 7. First, the main dimensions of the machine such as bore diameter and length are specified. Then the electric quantities and the main parameters of the machine

are calculated and determined. From thereon, the geometry of the lamination and the magnets is calculated, based on the magnetic circuit in the stator and the rotor lamination. The machine design is completed, specifying the winding quantities and the slot geometry. The design process of the machine is followed by a general design of the housing. Finally, the volume and mass of the housing and the machine are calculated and the volumetric and gravimetric power density are determined.

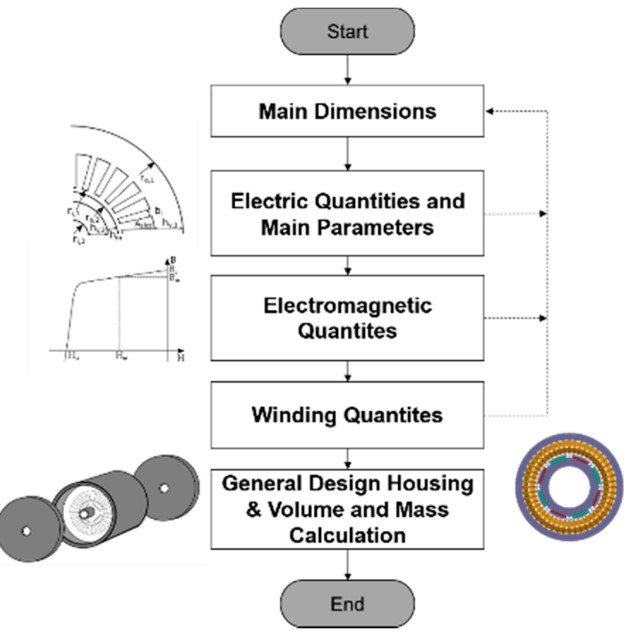

**Figure 7.** Design process of electric machines.

### 3.1.1. Main Dimensions

The first step of the design process is a first assessment of the main dimensions of the machine. The bore diameter and the length determine the provided torque of the machine. The torque

$$T = \frac{2\pi}{8} \, d_{i,1}^2 \, l_1 \hat{B}_{\mathrm{mp}} \hat{A}_p \tag{5}$$

is calculated using the spatial fundamental of the electric loading $\hat{A}_p$ and the spatial fundamental of the magnetic flux density of the magnet in the rotor $\hat{B}_{\mathrm{mp}}$ [18]. Estimating these two quantities, $\hat{A}_p$ and $\hat{B}_{\mathrm{mp}}$, the bore volume

$$\frac{2\pi}{8} \, d_{i,1}^2 \, l_i = \frac{P_{\mathrm{N}}}{2\pi \, n_{\mathrm{n}} \, \hat{B}_{\mathrm{mp}} \hat{A}_p} \tag{6}$$

can be calculated, if the rated power $P_{\mathrm{N}}$ and speed $n_{\mathrm{N}}$ are given. By setting the ratio of length and bore diameter to $\frac{l_i}{d_{i,1}} = 1.3$, the length and the diameter are determined. This ratio is set accordingly to other traction drives. The electric loading $\hat{A}_p$ and magnetic flux density $\hat{B}_{\mathrm{mp}}$ are chosen according to the maximum speed, (cf. Table 4). For higher speeds, the electric loading $\hat{A}_p$ is chosen smaller. The calculated lengths and diameters of the four machine designs are shown in Table 5.

The parameters need to be validated concerning their maximum surface velocity $v_{\mathrm{rot}}$. The PMSM-B1 and PMSM-B2 machine designs have interior magnets. Their surface velocity should not exceed $v_{\mathrm{rot,max}} = 120 \, \mathrm{m/s}$ [6]. The surface mounted magnets of PMSM-S1 and PMSM-S2 are kept by a bandage. The bandage leads to a bigger magnetic air gap, but allows to increase the maximum surface velocity to $v_{\mathrm{rot,max}} = 250 \, \mathrm{m/s}$ [5]. The air gap for the designs with interior magnets is assumed to be $\delta = 1 \, \mathrm{mm}$. For the PMSM-S1 surface mounted rotor, the bandage height is set to 2 mm and, for the PMSM-S2, the bandage height is set to 4 mm.

The surface velocity of the machine designs

$$v_{\text{rot}} = 1.1 \cdot \pi \cdot n_{\text{max}} \cdot d_{o,2} \tag{7}$$

is calculated for an overspeed of 10% of the maximum speed. The results for these machines are listed in Table 5 and do not exceed the limits of the surface velocity. The PMSM-B1 and PMSMS-S1 machine designs show low mechanical utilization compared to the limits. By increasing the bore radius and though the surface velocity, the machine would become larger and heavier.

**Table 5.** Main dimensions, electric loading and magnetic flux density of the magnet.

|  | PMSM-B1 | PMSM-B2 | PMSMS-S1 | PMSM-S2 |
|---|---|---|---|---|
| **Magnetic flux density of the magnet** $\hat{B}_{\text{mp}}$ | 0.95 T | 0.9 T | 0.8 T | 0.8 T |
| **Electric loading** $\hat{A}_p$ | 115 kA/m | 100 kA/m | 85 kA/m | 70 kA/m |
| **Core length** $l$ | 152 mm | 138 mm | 115 mm | 103 mm |
| **Bore diameter** $d_{i,1}$ | 117 mm | 106 mm | 88 mm | 79.5 mm |
| **Mechanical air gap** $\delta_{\text{m}}$ | 1 mm | 1 mm | 0.5 mm | 0.5 mm |
| **Magnetic air gap** $\delta$ | 1 mm | 1 mm | 2.5 mm | 4.5 mm |
| **Surface velocity** $v_{\text{rot}}$ | 72 m/s | 109 m/s | 151 m/s | 226 m/s |

### 3.1.2. Electric Quantities

To distinguish the rated current $I_N$ of the machine at rated power $P_N$, the apparent power

$$S_N = \frac{P_N}{\eta_N \, \cos(\phi_N)} \tag{8}$$

is calculated. The rated phase current

$$I_{N,\text{str}} = \frac{S_N}{m_1 \cdot U_{\text{str,max}}} \tag{9}$$

is then determined using the maximal phase voltage $U_{\text{str,max}}$ provided by the inverter. $m_1$ is the number of phases. To determine the current and the apparent power the efficiency $\eta_N$ and the power factor $\cos(\phi_N)$ at rated operation must be assumed reasonably. In this case $\eta_N$ and $\cos(\phi_N)$ are set to $\eta_N = 0.95$ and $\cos(\phi_N) = 0.9$. With the maximum phase voltage of $U_{\text{str,max}} = 230$ V, the rated current of $I_{N,\text{str}} = 127.67$ A is calculated. The maximum current deliverable by the inverter is $I_{\text{max,str}} = 205$ A.

The maximum operating frequency

$$f_{\text{max}} = \frac{n_{\text{max}}}{p} \tag{10}$$

is determined based on the number of pole pairs $p$. The number of pole pairs is kept small to keep the hysteresis and eddy current losses low. The number of stator slots $N_1$ is chosen to feature a high fundamental winding factor and a good compromise between reasonable slot dimensions and a low harmonic leakage factor. The resulting number of slots per pole and phase is accordingly set to $q = 2$. The pole pitch $\tau_p = \frac{d_{i,1}\pi}{2p}$ and the stator slot pitch $\tau_N = \frac{d_{i,1}\pi}{N_1}$ are then defined. The results are shown in Table 6.

**Table 6.** Electric quantities and main parameters.

|  | PMSM-B1 | PMSM-B2 | PMSMS-S1 | PMSM-S2 |
|---|---|---|---|---|
| **Number of pole pairs** $p$ | 4 | 3 | 2 | 2 |
| **Maximum frequency** $f_{\text{max}}$ | 0.88 kHz | 1.1 kHz | 1.1 kHz | 1.83 kHz |
| **Number of slots** $N_1$ | 48 | 36 | 24 | 24 |

### 3.1.3. Magnetic Quantities and Geometry of the Magnetic Circuit

The geometry of the stator and the rotor lamination, as well as the magnet geometry, are initially based on the estimation of the magnetic flux density in the air gap $\hat{B}_p$ at rated speed and rated power. This flux density is defined by the coupling between the stator and rotor flux and is accordingly higher than the flux density just due to the magnet, that was defined in Section 3.1.1. This effect is higher for the machines with interior magnets since their air gap is much smaller than for the machines with surface mounted magnets. The assumptions of the flux density for the four designs can be seen in Table 7. The magnetic flux per pole

$$\hat{\phi} = l \cdot \tau_p \cdot \frac{2}{\pi} \cdot \hat{B}_p \tag{11}$$

is calculated based on this assumption. By knowing the magnetic flux per pole, the tooth width and the height of the stator and the rotor yoke can be calculated by specifying the permissible magnetic flux density in the teeth $\hat{B}_t$, in the stator yoke $\hat{B}_{y,1}$ and in the rotor yoke $\hat{B}_{y,1}$ for rated operation. These values are set different for the four machine designs. The machine designs with the lower operating frequency $f$ can be operated with higher flux densities. This is due to eddy-current losses in the iron that depend on the flux density and the frequency. For higher speed, the permissible magnetic flux density should be set lower to limit the eddy-current losses. The chosen permissible magnetic flux is shown in Table 7. To determine the yoke height

$$h_{y,i} = \frac{\hat{\phi}}{\hat{B}_{y,i} \cdot l \cdot 2}, \tag{12}$$

the magnetic flux must be divided by the length of the machine $l$ the permitted magnetic flux density and a factor of 2, since the magnetic flux splits up in the two directions of the yoke [18]. The index i stands for either stator $i = 1$ or rotor $i = 2$.

The tooth width is calculated by multiplying the slot pitch with the ratio of the magnetic flux density in the air gap and the allowed magnetic flux density in the teeth as

$$w_{t,1} = \tau_N \frac{\hat{B}_{mp}}{\hat{B}_{t,1}}. \tag{13}$$

The results for the four machine designs are shown in Table 7.

**Table 7.** Magnetic quantities and geometry. Data from [18].

| | PMSM-B1 | PMSM-B2 | PMSMS-S1 | PMSM-S2 |
|---|---|---|---|---|
| **Magnetic flux density in the air gap $\hat{B}_p$** | 1.2 T | 1.1 T | 0.9 T | 0.9 T |
| **Magnetic flux density in the stator teeth $\hat{B}_{t,1}$** | 1.8 T | 1.8 T | 1.7 T | 1.6 T |
| **Magnetic flux density in the stator yoke $\hat{B}_{y,1}$** | 1.3 T | 1.2 T | 1.0 T | 0.9 T |
| **Magnetic flux density in the rotor yoke $\hat{B}_{y,2}$** | 1.4 T | 1.3 T | 1.1 T | 1.0 T |
| **Magnetic flux $\hat{\phi}$** | 5.1 mVs | 5.1 mVs | 4.3 mVs | 3.5 mVs |
| **Stator tooth width $w_{t,1}$** | 5.1 mm | 5.6 mm | 6.1 mm | 5.8 mm |
| **Stator yoke height $h_{y,1}$** | 13.5 mm | 16.2 mm | 19.9 mm | 19.8 mm |
| **Rotor yoke height $h_{y,2}$** | 12.5 mm | 14.9 mm | 18.1 mm | 17.9 mm |

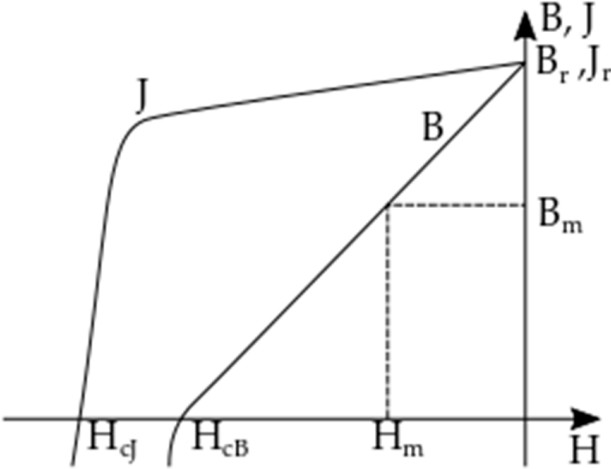

**Figure 8.** Schematic B-H and J-H diagram.

The size of the magnet, i.e., the width $w_m$ and the height $h_m$, are set according to the flux density in the air gap and the *B-H* diagram of the magnet (cf. Figure 8) [50]. The flux density of the magnet in rated operation is determined from the estimated flux density in the air gap by

$$B_m = \frac{\hat{B}_{mp}}{\frac{4}{\pi}}. \tag{14}$$

To take into account the drop of magnetic voltage in the iron, in comparison to the drop of voltage over the air gap, a saturation factor of $k_{sat} = \frac{V_{Fe}+V_\delta}{V_\delta}$ is defined and set to $k_{sat} = 1.6$. To take into account the slotting of the stator, the carter factor $k_c$ is used. $k_c$ is set to 1.1 [18]. The magnetic motive force of the magnet

$$\theta_m = k_{sat} \, k_c \, \delta \, \frac{B_m}{\mu_0} \tag{15}$$

is calculated regarding these phenomena. The magnetic field strength of the magnet $H_m$ is determined from the *B-H* diagram of the magnet, as depicted in Figure 9. The height of the magnet is then defined by the magnetic motive force $\theta_m$ and the determined magnetic field strength $H_m$

$$h_m = \frac{\theta_m}{H_m}. \tag{16}$$

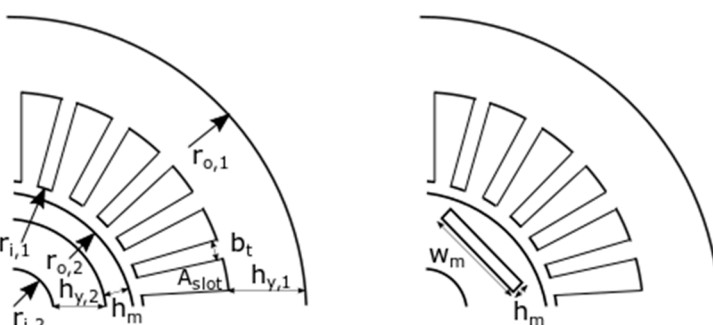

**Figure 9.** Machine geometry.

Since the machine shall be short-circuit-resistant, the height of the magnet needs to be validated after finalizing the winding configuration.

For a PMSM with interior magnets the width of the magnet

$$w_\mathrm{m} = \sin\left(\frac{\pi}{2\,p}\right) \cdot d_{\mathrm{i},1} \cdot 0.85 \tag{17}$$

is set to 85% of the side length of the equilateral triangle of one rotor pole pitch. For a PMSM with surface mounted magnets the width of the magnet is defined by the pole pitch

$$w_\mathrm{m} = \tau_p = \frac{\pi\, d_{\mathrm{i},1}}{2\,p}. \tag{18}$$

The bore diameter defines the magnetic flux in the machine and must be chosen accordingly.

Knowing the size of the magnet and the rotor yoke height, the rotor geometry can be completed. For a surface magnet rotor, the inner radius of the rotor is determined by

$$d_{i,2} = d_{o,2} - 2\,h_\mathrm{m} - 2\,h_{y,2} \tag{19}$$

If the magnets are interior in a bar shape, the depth is considered by multiplying the magnet height by a factor $\beta$ to determine the inner radius:

$$d_{i,2} = d_{o,2} - 2\,\beta\,h_\mathrm{m} - 2\,h_{y,2}. \tag{20}$$

The factor is set to $\beta = 3$ for the machines with interior magnets. The magnet parameters are shown in Table 8.

**Table 8.** Magnetic quantities.

| | PMSM-B1 | PMSM-B2 | PMSMS-S1 | PMSM-S2 |
|---|---|---|---|---|
| Magnet material | | Sm2Co17 | | |
| Coercive field strength $H_{\mathrm{cJ}}$ | | 2000 kA/m | | |
| Remanence $B_\mathrm{r}$ | | 1.13 T | | |
| Field strength of the magnet at operating point $H_\mathrm{m}$ | 280 kA/m | 300 kA/m | 380 kA/m | 380 kA/m |
| Magnetic flux density of the magnet at operating point $B_\mathrm{m}$ | 0.746 T | 0.707 T | 0.628 T | 0.628 T |
| Magnetic motive force $\theta_\mathrm{m}$ | 1045 kA | 990 kA | 2200 kA | 3960 kA |
| Height of the magnet $h_\mathrm{m}$ | 3.7 mm | 3.3 mm | 5.8 mm | 10.4 mm |
| Width of the magnet $w_\mathrm{m}$ | 37.4 mm | 44.3 mm | 69.3 mm | 62.4 mm |
| Inner rotor diameter $d_{\mathrm{i},2}$ | 67.6 mm | 54.5 mm | 35.5 mm | 13.8 mm |

### 3.1.4. Winding Quantities

The maximum number of series turns per phase is determined for the operating point at rated speed $n_\mathrm{N}$ and rated power $P_\mathrm{N}$. The magnetic flux per pole $\hat{\phi}$ at this point has already been estimated with the magnetic flux density in the air gap (see Equation (9)). Assuming the winding factor $\xi_p = 0.92$ the maximum number of turns per phase

$$w_\mathrm{calc} = \frac{\sqrt{2}U_\mathrm{str,max}}{2\pi \cdot n_\mathrm{N} \cdot p \cdot \xi_p \cdot \hat{\phi}} \tag{21}$$

can be calculated [18]. The number of turns can be achieved with different winding configurations. The number of conductors per slot $z$, the number of slots per pole and phase $q$ and the number of parallel branches $a$ determine the winding configuration and the winding layout. The number of series turns per phase is set by these winding parameters to

$$w = \frac{z \cdot q \cdot p}{a} \tag{22}$$

and should be close to the value determined. The chosen values for the winding quantities are listed in Table 9. This winding configuration also determines the previously estimated spatial fundamental of the electric loading, calculated by

$$\hat{A}_p = \sqrt{2}\,\xi_p \frac{m\,w\,I_{\text{N,str}}}{\pi \frac{d_{i,1}}{2}}. \tag{23}$$

After setting the winding parameters, the geometry of the slot can be determined. The cross-sectional area of the conductors

$$A_{\text{co}} = \frac{I_{\text{N,str}}}{S_{1,\text{cont}}} \tag{24}$$

is calculated from the rated current $I_{\text{N,str}}$ and the current density $S_{1,\text{cont}}$. The machines are supposed to have a water jacket. The current density is accordingly assumed at $S_{1,\text{cont}} = 12\,\text{A/mm}^2$.

Depending on the type of winding and the manufacturing process, the copper fill factor $k_{\text{co}}$ must be estimated. In this case for an automated round wire winding, it is set to $k_{\text{co}} = 0.36$ [50]. The area of the slot is then determined by

$$A_{\text{slot}} = z\frac{A_{\text{co}}}{k_{\text{co}}}. \tag{25}$$

In the case of a round wire winding, the slot can be designed as a trapezium. The width of the slot at the bore radius is set by the width of the tooth and the bore radius to

$$b_{\text{slot, i}} = \frac{\pi\,d_{i,1}}{N_1} - b_t. \tag{26}$$

The width of the slot at the bottom of the slot $b_{\text{slot,u}}$ and the corresponding slot height $h_{\text{slot}}$ are calculated with the relation

$$h_{slot} = \frac{2 \cdot A_{\text{slot}}}{(b_{slot,\,l} + b_{slot,u})} \tag{27}$$

and the width of the trapezium at the bottom of the slot with

$$b_{\text{slot,o}} = \frac{2\pi \cdot r_{i,1} + 2\,h_{\text{slot}}}{N_1} - b_t. \tag{28}$$

Using the height of the slot, the outer radius of the stator is calculated as

$$d_{o,1} = d_{i,1} + 2\,h_{\text{slot}} + 2\,h_{1,y}. \tag{29}$$

**Table 9.** Winding configuration.

|  | PMSM-B1 | PMSM-B2 | PMSMS-S1 | PMSM-S2 |
|---|---|---|---|---|
| Calculated number of series turns per phase $w_{\textbf{calc}}$ | 41.37 | 38.56 | 27.8 | 21.57 |
| Number of conductors per slot $z$ | 20 | 16 | 12 | 10 |
| Number of slots per pole and phase $q$ | 2 | 2 | 2 | 2 |
| Number of parallel branches $a$ | 4 | 3 | 2 | 2 |
| Final number of series turns per phase $w$ | 40 | 32 | 24 | 20 |
| Electric loading $\hat{A}_p$ | 108.39 kA/m | 95.56 kA/m | 89.71 kA/m | 79.88 kA/m |
| Conductor cross sectional area $A_{\text{co}}$ | 2.66 mm$^2$ | 3.55 mm$^2$ | 5.32 mm$^2$ | 5.32 mm$^2$ |
| Slot area $A_{\text{slot}}$ | 147.8 mm$^2$ | 157.6 mm$^2$ | 177.3 mm$^2$ | 147.8 mm$^2$ |
| Inner slot width $b_{\text{slot,i}}$ | 2.5 mm | 3.6 mm | 5.4 mm | 4.5 mm |
| Outer slot width $b_{\text{slot,o}}$ | 6.7 mm | 8.2 mm | 10.9 mm | 10.1 mm |
| Slot height $h_{\text{slot}}$ | 31.8 mm | 26.2 mm | 22 mm | 21.1 mm |
| Stator outer diameter $d_{o,1}$ | 208 mm | 191 mm | 170 mm | 161 mm |

Finally, the design needs to be validated to determine if it is short-circuit-resistant. This is done by calculating the magnetic motive force

$$\hat{\theta}_{sc} = 4\pi\frac{m}{2}4\ \sqrt{2}\ I_{N,str}\ w\frac{\xi_w}{2p}$$ (30)

in case of a sudden short circuit from rated operation occurs [51]. The calculation is based on the maximum short-circuit current. The winding attempts to keep the magnetic flux constant while the rotor keeps rotating. The short-circuit current is assumed to be 4 times the current at rated power. In the worst case, the magnetic flux is directed opposite to the magnetization direction of the magnet. The magnetic field strength of the winding, in case of the sudden short circuit, leads to a decrease of the magnetic polarization $J$ of the magnet (cf. Figure 9). If the magnetic motive force of the winding is higher than the product of coercive field strength $H_{cJ}$ and the height of the magnet $h_m$, the magnet will be demagnetized, leading to the requirement

$$h_m \geq \frac{\theta_{sc}}{H_{cJ}}.$$ (31)

The resulting heights of the magnets are listed in Table 10 for the four machine designs. All machine designs are short circuit resistant.

**Table 10.** Short circuit resistant magnet height.

|  | **PMSM-B1** | **PMSM-B2** | **PMSMS-S1** | **PMSM-S2** |
|---|---|---|---|---|
| **Height of the magnet $h_m$** | 3.2 mm | 3.4 mm | 3.8 mm | 3.2 mm |

The active parts, i.e., rotor and stator lamination, winding and magnets, of the machine designs are shown in Figure 10. It can be noticed, that the machine size decreases with increased maximum speed, as expected.

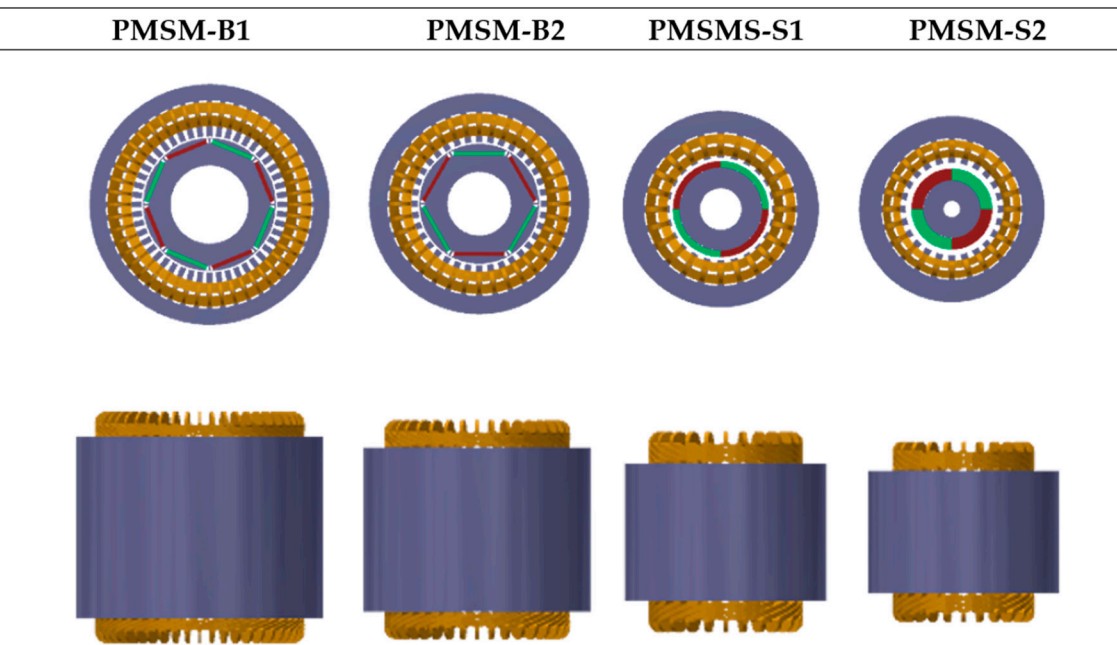

**Figure 10.** Final design of PMSM-B1, PMSM-B2, PMSM-S1 and PMSM-S2.

### 3.2. Housing Design

The housing consists of four components as depicted in Figure 11: The shaft, the water jacket and the two end shields. The components are designed as simple geometric bodies with dimensions set according to the machine designs.

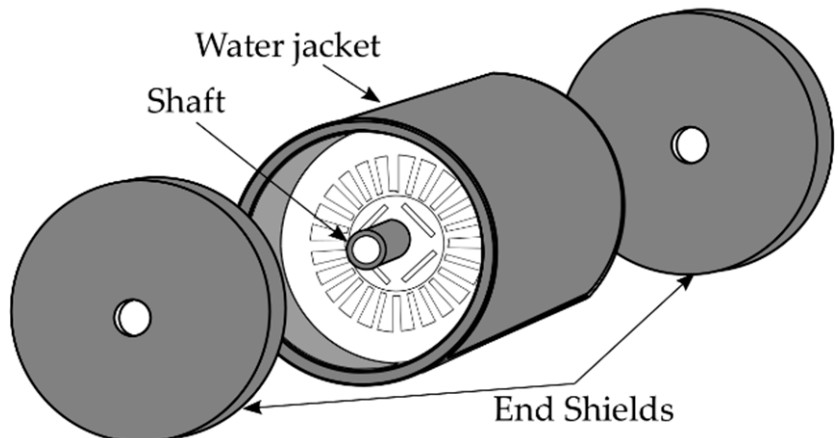

**Figure 11.** Housing components.

The shaft and the water jacket are approximated as hollow cylinders. The inner diameter of the shaft is set to

$$d_{\text{i,shaft}} = \frac{d_{\text{i},2}}{1.8}. \tag{32}$$

The outer diameter is equal to the inner diameter of the rotor. The inner diameter of the water jacket is set to the outer diameter of the stator and the outer diameter is approximated by

$$d_{\text{o,c}} = d_{o,1} \cdot 1.1. \tag{33}$$

The length of the housing

$$l_{\text{housing}} = 1.5 \cdot l_1 \tag{34}$$

includes space for the end windings and the winding protection of the machine. The housing is closed with the end shields, that are approximated as cylinders, on both sides. The outer diameter of the end shields is equal to the outer diameter of the water jacket and the height of the end shields is set to 20 mm.

### 3.3. Electric Machine Design Results

Finally, the volume and the mass of the housing and the machine are calculated. The volume of the components is determined based on the derived geometric quantities. The weight is calculated based on material densities, shown in Table 11.

**Table 11.** Volume of the machine designs and volumetric power density.

| | Rotor and Stator | Winding | Winding protection | Magnet | Housing | Shaft |
|---|---|---|---|---|---|---|
| **Material** | Electric steel | Copper | Epoxy | Sm2Co17 | Aluminum | Construction steel |
| **Density** | 7600 kg/m$^3$ | 8960 kg/m$^3$ | 1850 kg/m$^3$ | 8400 kg/m$^3$ | 2700 kg/m$^3$ | 7720 kg/m$^3$ |

For the housing a weight of 60% of the housing weight is added, to consider additional components, i.e., bearings, retaining rings and screws. To calculate the volume of the winding and the winding protection, the end windings need to be considered. For the PMSM-S1 and the PMSM-S2 machine designs, the volume of the end winding and winding protection is assumed to be the same size, as in the active part. For the machine design PMSM-B1 and PMSM-B2, it is assumed to be 60% of the active part. The machine designs PMSM-S1 and PMSM-S2 have a larger end winding, due to the smaller number of pole pairs $p$.

The weight, the volume and the volumetric and gravimetric power density, in case of maximum power, of the four designs are listed in Table 12. The results are discussed in Section 5, together with the results of the transmission.

**Table 12.** Volume, mass and volumetric and gravimetric power density of the machine designs.

|  | PMSM-B1 | PMSM-B2 | PMSMS-S1 | PMSM-S2 |
|---|---|---|---|---|
| **Volume $V$** | 7.5 dm$^3$ | 5.8 dm$^3$ | 4.3 dm$^3$ | 3.45 dm$^3$ |
| **Weight $m$** | 44.1 kg | 34 kg | 23.6 kg | 18.6 kg |
| **Volumetric power density $p_v$** | 17.9 kW/dm$^3$ | 23.1 kW/dm$^3$ | 31.3 kW/dm$^3$ | 39 kW/dm$^3$ |
| **Gravimetric power density $p_g$** | 3.05 kW/kg | 3.97 kW/kg | 5.72 kW/kg | 7.22 kW/kg |

## 4. Conceptual Design of High-Speed Transmissions

After the introduction of the design process for the high-speed electric machines in Section 3, this section will give an overview of the design process for the associated high-speed transmissions. Like the electric machines, the transmissions are designed to match the reference vehicle's requirements for a rated power of 75 kW (cf. Table 1). In order to obtain comparable transmission designs, all systems feature the same speed and torque on the drive axle. The workflow in Figure 12 shows the design process for the transmission with the desired maximum input speed of the electric machine. After selecting the maximum speed, the necessary overall gear ratio follows directly by considering the data of the reference vehicle. The next step is the selection of the desired transmission configuration. This study considers two- and three-stage helical gear units and a combination of a planetary and helical gear set in axial parallel configurations. In the next steps, the wheel set of the transmission, including the gearings, wheel bodies, shafts and bearings is generated with the software GAP [52]. This program is able to generate transmission designs automatically and can additionally optimize the structure with regard to a large number of target variables [53]. In order to design the gearing, practical gear parameters are specified as optimization goals. To ensure comparability of the gearings, uniform target values were defined for safety factors in the load capacity calculation. The actual design of the toothing is carried out with the software STplus [54] in accordance with DIN ISO 6336 [55]. GAP also contains a structure generator for the rapid design of wheel bodies, shafts and bearings. After using the GAP structure generator, a draft of the wheel set is available that can be used to generate a model of the transmission housing. Finally, this process results in a complete concept of a transmission based on the selected configuration and overall gear ratio suitable for the reference vehicle. The following subsection will explain the individual steps of the design process in more detail.

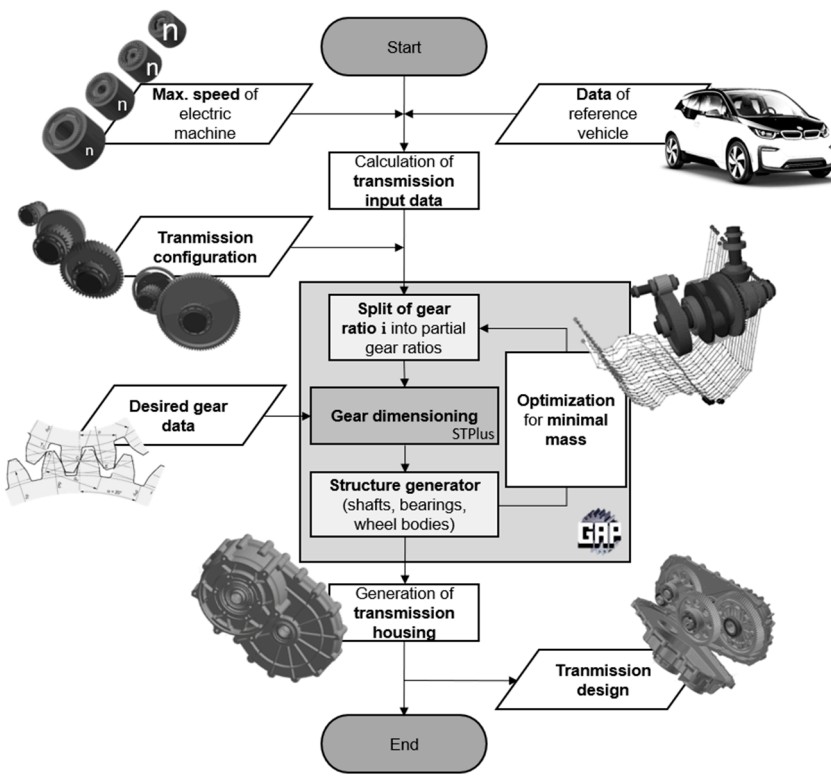

**Figure 12.** Transmission design process.

### 4.1. Calculation of Transmission Input Data

The first step of the design process is the calculation of the required input data (power, speed and torque) of the transmission. In order to obtain comparable powertrains for the reference vehicle, the transmission output parameters (see Section 1) on the drive axle are kept constant. With the maximum speed on the drive axle $n_{max,out}$ and the maximum speed of the considered electric machine $n_{max}$, the required overall gear ratio $i$ follows.

$$i = \frac{n_{max}}{n_{max,out}}. \tag{35}$$

The overall gear ratio $i$ of the transmission together with the known maximum torque of the reference vehicle on the drive axle $T_{max,out}$ lead to the maximum torque at the transmission input.

$$T_{max} = \frac{T_{max,out}}{i}. \tag{36}$$

In addition to the peak parameters at the transmission input, the rated input parameters are important for the design process. The rated $P_n$ power of the electric machines is 75 kW and can be used to calculate the rated torque of each transmission.

$$T_n = \frac{P_N}{2 \cdot \pi \cdot n_N}. \tag{37}$$

Using these simple formulas, the required gear ratio and input torque can be determined for each considered input speed of the transmission.

### 4.2. Transmission Configurations

In the course of this study, three different transmission layouts in axial parallel configurations are considered. Figure 13 shows the transmission configurations, which are from left to right a two- (2ST)

and three-stage (3ST) helical gear unit and a combination of a planetary and helical gear set (PLST). Helical gear sets with one stage are not considered, since the high gear ratios, which are considered in this study, lead to very high volumes. In case of the PLST, the first stage is the planetary gear set, since a planetary gear set on the drive axle would lead to complicated designs. Due to the small rotor diameter for higher speeds, a coaxial transmission configuration with a drive shaft guided through the rotor is not considered, as there would not be enough installation space.

| 2 ST | 3ST | PLST |
| --- | --- | --- |

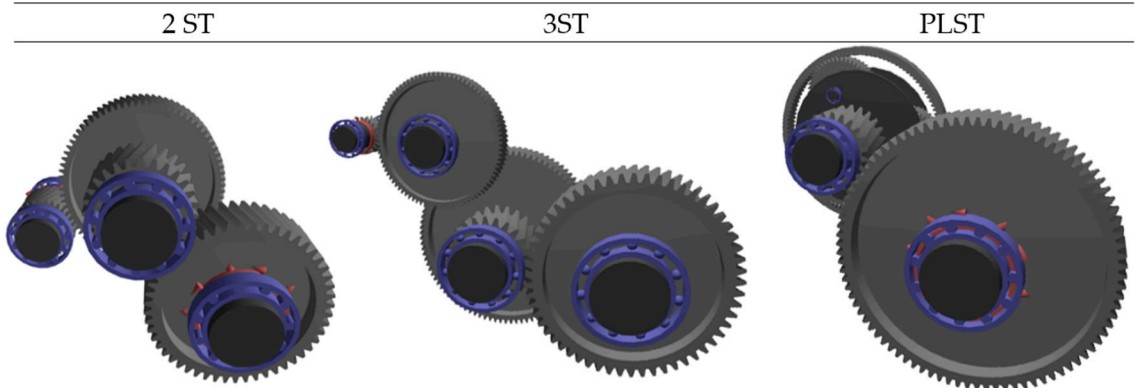

**Figure 13.** Considered transmission configurations.

*4.3. Transmission Design with GAP*

After the selection of the transmission configuration, the next step is to design the transmission system using the GAP software. The software is suitable for the rapid generation of transmission designs including the gearings, shaft dimensioning and calculation of the bearing loads. It is possible to define target functions such as minimal mass, volume or maximum efficiency and load capacity to optimize the design [56]. Since the study focuses on the power density of the powertrain, the transmissions are optimized to reach minimal mass. Therefore, the first step of the actual design process is the splitting of the overall gear ratio *i* into the transmissions' partial gear ratios.

4.3.1. Determination of the Transmissions Partial Gear Ratios

The overall gear ratio *i* of a transmission is generally made up of the transmission's stage gear ratios $i_k$, which are in series. Bansemir [57] developed an approach with a parameterized power function for determining the stage gear ratios of the $k_{max}$ stages, taking into account an exemplary shaft mass.

$$i_k = a_k \cdot i^{b_k} \qquad \text{for } k < k_{max} \tag{38}$$

$$i_{k_{max}} = \frac{i}{\prod_{k=1}^{k_{max}-1} i_k} \tag{39}$$

The index *k* identifies the current number of the considered stage and $k_{max}$ accordingly for the number of stages of the transmission. The parameters $a_k$ and $b_k$, which control the splitting of the overall gear ratio into the stage gear ratios, are the result of an optimization for each transmission configuration and will be shown in Section 4.4.

4.3.2. Gear Dimensioning and Design

With the partial gear ratios of the transmission, the actual design of the gearings are calculated with the STplus software [54]. This software includes the load capacity calculation according to DIN ISO 6336 [55] and additional content regarding the geometrical design and machining of spur gear stages. The starting point of the design process are uniform values of the safety factors for all transmissions

that lead to comparable gearing designs regarding the load carrying capacity. Table 13 shows the minimal safety factors for the load carrying capacity calculation. $S_{F,min}$ stands for the minimal safety factor against tooth root breakage, $S_{H,min}$ against pitting and $S_{B,min}$ represents the minimal required safety against scuffing. In addition to the safety factor that should be achieved, the considered load cases play an important role in gear dimensioning.

**Table 13.** Safety factors for the load carrying capacity calculation.

| Safety Factors | |
|---|---|
| $S_{F,min}$ | 1.3 |
| $S_{H,min}$ | 1.1 |
| $S_{B,min}$ | 1.3 |

Table 14 gives an overview of the 5 load cases for the gear design process. Load case 1 is a load spectrum for a middle-class BEV, which has a total operating time of 2500 h and covers the map of the electric machine by scaling the torque and speed according to their rated and maximum operating points. In addition to the load collective, the overload case (2) covers an operation of the system with maximum torque and an application factor of $K_A = 1.6$. The possible operating time of this case should be at least 3 h. Load case 3 represents a static overload due to inappropriate utilization with maximum torque and an application factor of $K_A = 2$. Load cases 4 and 5 ensure the scuffing load capacity, whereby critical operating points with high-speed and load (4) and maximum load at rated torque (5) are taken into account.

**Table 14.** Load cases for the gearing design.

| Load Case | Input Torque T | Input Speed n | Operating Time |
|---|---|---|---|
| 1—Load spectrum | $0 \ldots T_{max}$ | $0 \ldots n_{max}$ | 2500 h (200,000 km) |
| 2—Overload | $T_{max} \cdot 1.6$ | Rated speed $n_N$ | >3 h |
| 3—Static overload | $T_{max} \cdot 2$ | - | Short term |
| 4—Scuffing | $T_{max}$ at $n_{max}$ | $n_{max}$ | Short term |
| 5—Scuffing | $T_{max}$ | $n_N$ | Short term |

To receive comparable gearing designs, uniform values for central gear parameters and other design parameters are also specified. Table 15 shows important design factors according to DIN ISO 6336 [55] and design targets for the gear design optimization. The dynamic factor $K_V$ considers additional forces in the tooth contact due to dynamic effects. The load factors $K_{H\alpha}$ and $K_{H\beta}$ represent the influence of an uneven load distribution over the profile and width of the flank. Furthermore, in order to control the gear design process, design targets for the contact ratios $\varepsilon_\alpha$ and $\varepsilon_\beta$ as well as the width-to-diameter ratio b/d of the wheel bodies are defined. The selected design values are suitable for the practical preliminary design of gearings, but would have to be determined more precisely with higher-quality methods for detailed designs.

**Table 15.** Design factors and design targets for the gear design process.

| Design Factors | |
|---|---|
| Dynamic factor $K_V$ | 1.2 |
| Transverse load factor $K_{H\alpha}$ | 1.1 |
| Face load factor $K_{H\beta}$ | 1.2 |
| **Design targets** | |
| Transverse contact ratio $\varepsilon_\alpha$ | 1.6 |
| Overlap ratio $\varepsilon_\beta$ | 2.1 |
| Aspect ratio b/d | < 0.7 |

4.3.3. Structure Generator

To complete the transmission wheel set, the shafts, wheel bodies and bearing are generated in the next step. The shaft length follows from the numbers of bearings and wheel bodies, which are located on the shaft. Each element on the shaft is associated with an individual section. For these sections, an equivalent stress according to Niemann et al. [42] is calculated from the torsional and bending moments $T$ and $T_b$ to define the minimum shaft diameter $d_{V,a,min}$(Equation (38). The allowable stress $\sigma_{b,all}$ is derived from the fatigue strength $\sigma_w$ of the material, which is 20MnCrS5 in case of the shafts.

$$d_{V,a,min} = 2.17 \cdot \sqrt[3]{\frac{T \cdot 1000 \cdot \sqrt{a_b^2 + 0.4}}{\sigma_{b,all}}} \text{ with } a_b = \frac{T_b}{T} \text{ and } \sigma_{b,all} = \frac{\sigma_w}{S} \tag{40}$$

Furthermore, Parlow's [53] approach for designing a hollow shaft is used, which calculates the minimum outer diameter $d_{HW,a,min}$ and maximum inner diameter $d_{HW,i,min}$ to achieve the identical moment of resistance as the solid shaft. While the shafts of the spur gear sets are solid shafts, the design of the planetary gear set requires a hollow shaft for the planet carrier.

Figure 14 illustrates the simplified approach to determine the length of the transmissions' shafts. When calculating the shaft length, the input variables are the tooth widths $b_w$, the distances between the wheel bodies $d_{ww}$, the distances between the wheel bodies and bearings $d_{wb}$ and bearing widths $b_b$. The distances are fixed at a value of $d_{wb} = d_{ww} = 10$ mm, the toothing widths $b_w$ are a result of the gear dimensioning and the bearing widths are changed depending on the torque at the specific shaft. Arrangements with a locating and a non- locating bearing, using a cylindrical roller and a deep groove ball bearing are used for all shafts. Since the loads on the output shafts are very similar for all transmission designs, a uniform bearing concept was chosen. The bearing sizes and weights of the remaining bearings were scaled from the design with the maximum input speed according to the torque and speed.

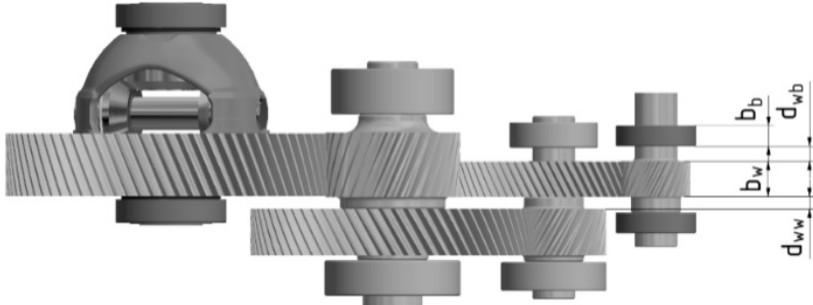

**Figure 14.** Schematic structure for definition of the shaft length.

Finally, the wheel body structures are calculated in the structure generator. Figure 15 shows the geometry of the wheel bodies. The geometry depends on the root diameter $d_f$ and the tooth width $b$ parameters, where the web width is $b_s = 0.25 \cdot b$. The thickness of the gear rim follows the recommendations of the ISO6336 standard, in which a minimum ring gear thickness of three times the module $m_n$ is required. Therefore, the diameter results in $d_{rim} = d_f - 6 \cdot m_n$. The hub geometry of the wheel body is defined by the inner and outer hub diameters $d_{ih}$ and $d_{oh}$ and the hub width, which is the tooth width. The inner hub diameter is equal to the outer shaft diameter, and the wheel hubs have a thickness of three times the module according to the procedure in case of the gear rim.

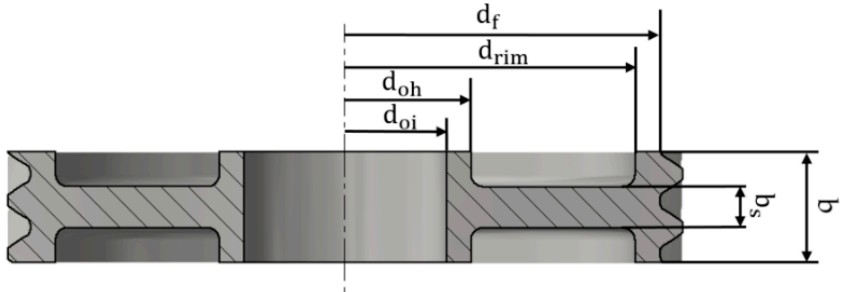

**Figure 15.** Schematic geometry of the wheel bodies.

### 4.3.4. Optimization of Transmission Design for Minimal Mass

With the conclusion of the structure generator, a first draft of a complete wheel set for the considered configuration and gear ratio is available. Since this study primarily looks at the power density of the system, the next step is to optimize the transmission to find an optimal design regarding systems minimal mass. The optimization used in the study is based on the simulated annealing algorithm [53,58], which is part of GAP and provides values for the parameters $a_k$ and $b_k$; this is introduced in Section 4.4 for the splitting of the gear ratio. Table 16 shows the optimized factors in case of the three considered transmission configurations. In case of the PLST configuration, an additional requirement for a sufficient center distance was introduced to ensure the installation space for the drive axle.

**Table 16.** Parameters for splitting of overall gear ratio to reach minimal mass.

| Transmission Configuration | $a_k$ | $b_k$ |
| :---: | :---: | :---: |
| 2ST | $a_1 = 1.0151$ | $b_1 = 0.6011$ |
| 3ST | $a_1 = 1.0151$ | $b_1 = 0.6240$ |
| | $a_2 = 1.7350$ | $b_2 = 0.2236$ |
| PLST | $a_1 = 0.4488$ | $b_1 = 0.6859$ |

### 4.3.5. Generation of Simplified Housing Geometry

Analogous to the design of the electric machines, the housing structure is also considered for the transmissions. To do this, an envelope curve is placed around the wheelset, the shaft angles being adjusted to reach a minimum base area of the housing. Figure 16 shows the base area of the considered three-stage configuration 3ST, which can be described by simple geometric shapes (circular sectors and trapezoid). When calculating the housing volume, a distance of at least 10 mm from all rotating components to the housing wall and a constant wall thickness of 5 mm is assumed.

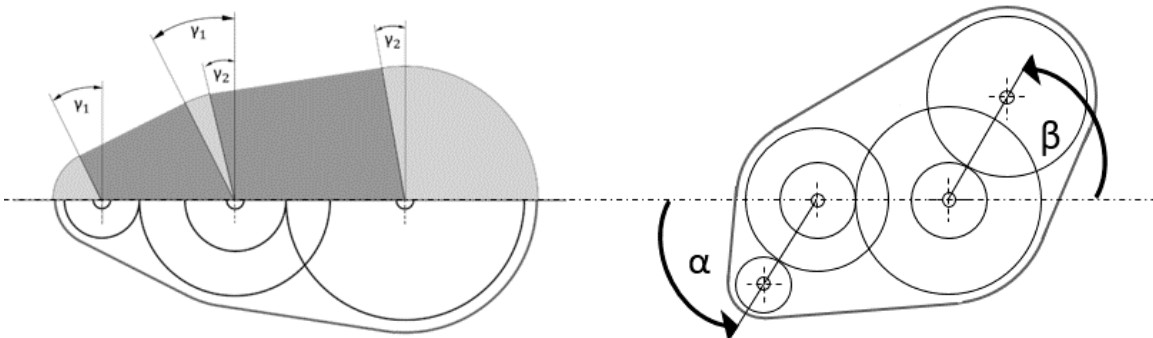

**Figure 16.** Exemplarily generation of a simplified housing geometry for 3ST.

Considering the maximum axis angles without collision of wheel bodies, the shaft angles α and β were optimized for each transmission design to achieve a minimum base area of the housing and thus a minimum volume and mass. A common transmission-housing alloy, AlSi9, with a density of $\rho_{AlSi9} = 2.65\ \text{kg/dm}^3$ was assumed as material. In addition, stiffening structures and variable wall thicknesses of the housing are considered according to Linke [59] by a correction factor of $f_h = 1.5$, which is multiplied with the housing mass derived from the described optimization.

### 4.4. Transmission Design Results

With the designed wheelset and the simplified housing geometry, the transmission design with the desired maximum input speed and gear ratio are finalized. An overview of the results of the masses and volumes of the designed transmissions is given in this section. Figure 17 shows the transmission masses of the three configurations and total gear ratios from 5 to 50 with the associated maximum speed of the electric machines. The PLST configuration shows the lowest transmission mass over the entire range of input speeds and gear ratios, whereby the mass increases with increasing gear ratios. As expected, ST3 has the largest overall transmission mass due to the additional shaft and wheel bodies, whereby a decreasing transmission mass with increasing gear ratio can be found. The reason for the decreasing mass in case of ST3 is the decreasing torque with increasing maximum speed of the electric machine and thus a smaller possible size of the gears and shafts. The ST2 configuration is between PLST and ST3, whereby ST2 becomes heavier than ST3 for gear ratios larger than i = 40.

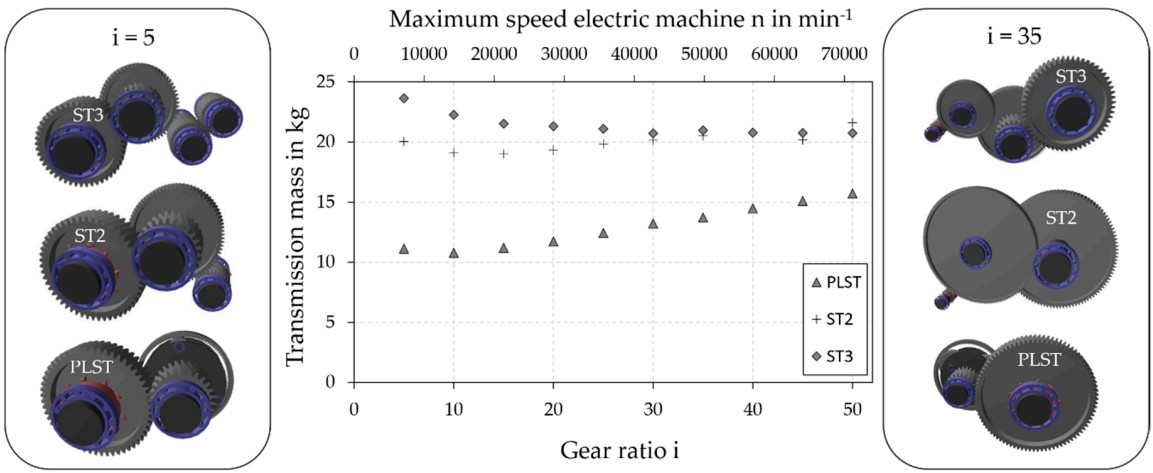

**Figure 17.** Transmission masses for the three considered configurations over the gear ratio.

Figure 18 shows the resulting volumes of the three transmission configurations. Again, the PLST configuration achieves the lowest values for the entire values of the volumes for the entire evaluation range. In case of the two-stage ST2 configuration, the large required wheel diameters quickly lead to a large overall system volume. The three-stage ST3 configuration shows the smallest increase in volume with increasing gear ratios and its total is in between the compact PLST and larger ST2 configuration.

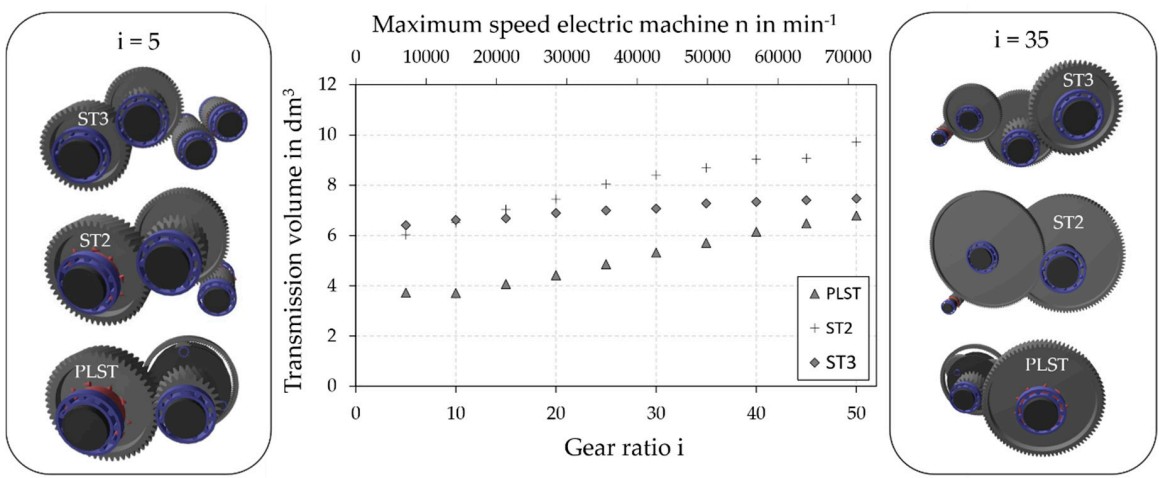

**Figure 18.** Transmission volume for the three considered configurations over the gear ratio.

## 5. Merged Design of High-Speed Powertrain

In this chapter, the results of the electric machine and transmission designs from Sections 3 and 4 are combined and presented. A total of four different electric machines with various maximum speeds but identical power values were designed, as described in Section 3. Analogously, suitable transmissions were designed in Section 4, which adapt the maximum speed of the electric machine with constant output parameters by adjusting the total gear ratio. In order to make a statement about characteristics of the whole powertrain, the transmissions that exactly match the designed electric machines were selected and combined. Through this procedure, twelve possible powertrain systems for the BMW i3 reference vehicle (cf. Chapter 1) will be presented, which differ in maximum speeds of the electric machine and transmission design. Based on these powertrain designs, the system is examined for total mass and volume and thus the power density of the electromechanical powertrain.

Figure 19 (left) shows the masses of the resulting powertrain designs over the gear ratio, respectively the maximum speed of the electric machines. The total masses show a clear decrease in powertrain mass for all transmission configurations. Since PLST achieves the lowest transmission masses, the powertrain system with this transmission configuration has the lowest overall mass, too. The increase of the maximum speed of the electric machines from 12,000 to 50,000 $\text{min}^{-1}$ reduces the weight for the PLST configuration from 55.0 to 32.4 kg, which is 22.6 kg or a mass reduction of 41.1%. In case of the two-stage 2ST configuration, the mass reduction is 24.2 kg from 63.4 to 39.2 kg or 38.2%, whereas for the three-stage 3ST configuration this is 27.1 kg from 66.7 to 39.6 kg or 40.7%. The weight of the electric machine is reduced by 57.6% from 44.1 to 18.7 kg by increasing the maximum speed of the electric machine. The diagram on the right in Figure 19 further clarifies the composition of the total mass from the electric machine and the transmission for the PLST configuration. It shows a significantly increasing share of the transmission mass in the total mass of the powertrain due to the rapidly decreasing weight of the electric machine. However, the transmission mass is not increased by the same extent as the weight of the electric machine is decreased, which makes it possible to strongly reduce the overall weight of the powertrain.

In addition to the powertrain design of this study, the diagram in Figure 20 (left) also shows the powertrain mass of the reference vehicle including the electric machine and transmission. The transmission of the reference vehicle has a gear ratio of 9.665 and a mass of 23.2 kg. Together with the electric machine, the reference powertrain achieves a total mass of 73.1 kg [60]. This value is close to the data point of this study with a gear ratio of 8.2.

Figure 20 shows similarly to Figure 19 the volumes of the powertrain designs with the four electric machines and the three different transmission configurations. Here, the increase of maximum speed from 12,000 to 50,000 $\text{min}^{-1}$ reduces the volume for the PLST configuration from 11.3 $\text{dm}^3$ to 9.2 $\text{dm}^3$,

which is 2.09 dm$^3$ or a reduction of 18.5%. In case of the two-stage 2ST configuration, the volume reduction results in 1.8 dm$^3$ from 14.0 dm$^3$ to 12.2 dm$^3$ or 13.1%. The largest volume reduction can be found for the ST3 configuration with 3.3 dm$^3$ from 14.1 dm$^3$ to 10.8 dm$^3$ or 23.6%. The volume of the electric machines decreases equivalently as the mass.

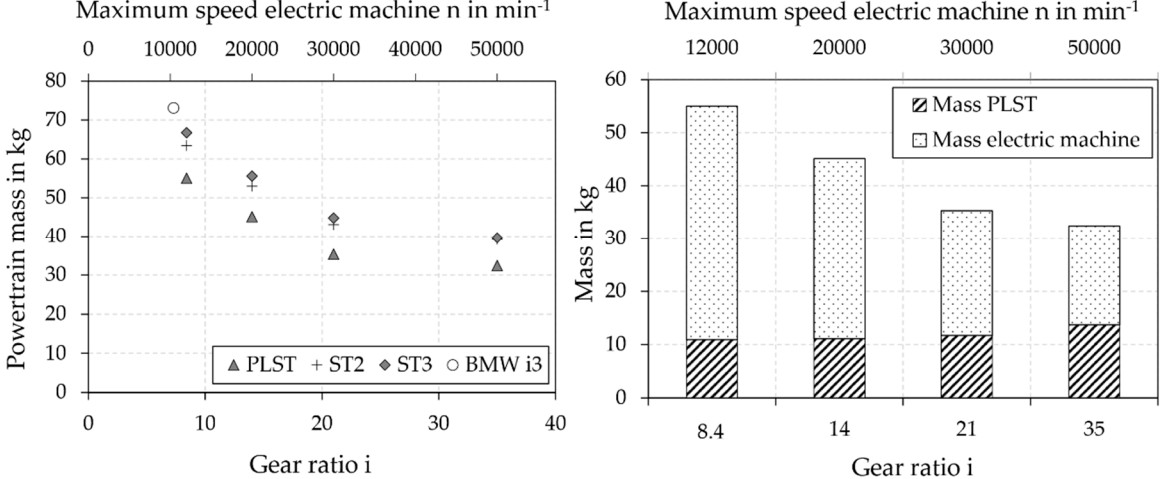

**Figure 19.** Powertrain masses (**left**) and its split into transmission and electric machine (**right**).

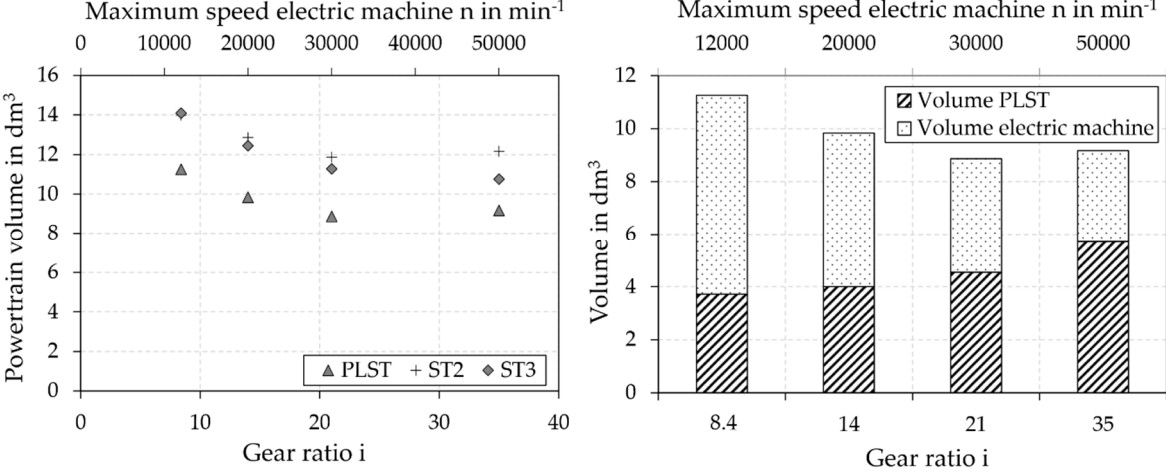

**Figure 20.** Powertrain volumes (**left**) and its split into transmission and electric machine (**right**).

In contrast to the total mass, the increase in the maximum speed from 30,000 to 50,000 min$^{-1}$ in the case of the PLST and ST2 configurations already shows an increasing total volume. The volume of these transmission configurations increases to a greater extent than the electric machine decreases its volume because of the large wheel diameters. While the share of the transmission mass for PLST in the total mass at 50,000 min$^{-1}$ is 42.3% (cf. Figure 20), the share for the volume is 62.3%, which is higher than the electric machine volume.

Figure 21 (left) shows the gravimetric power density of the powertrain designs and an increasing power density with increasing maximum speed of the electric machines in all three transmission configurations. Since PLST achieves the lowest masses, this configuration achieves the highest power density. The increasing difference between the PLST, ST2 and ST3 configurations, respectively with increasing gear ratio is due to the increasing share of the transmission mass to the total mass. In addition, the gravimetric power density of the reference vehicle's powertrain of 1.85 kW/kg is supplemented into the diagram. In comparison with the PLST configuration at maximum speed of 50,000 min$^{-1}$, the gravimetric power density can be increased by 125.7% to 4.17 kW/kg.

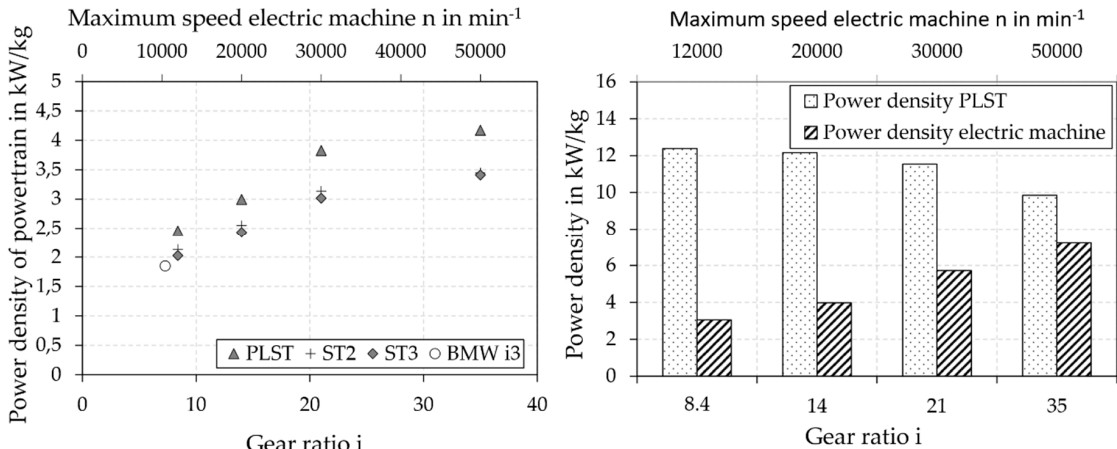

**Figure 21.** Gravimetric power density of powertrain (**left**), electric machine and transmission (**right**).

Figure 21 (right) shows the evolution of the gravimetric power density of the electric machine and the transmission PLST with increasing gear ratio. While there is a large gap between the very high value for the transmission and the low value of the power density of the electric machine at a low gear ratio, this gap is becoming lower and the values converge with increasing gear ratio.

When looking at the results of the volumetric power density in Figure 22, the effect of the disproportionately increasing transmission volume becomes apparent. Even at a gear ratio of 21, which equals the maximum speed of 30,000 $min^{-1}$, the electric machine has a higher volumetric power density than the transmission and expands the distance to the transmission with further increase of the maximum speed. Like in the case of the powertrain mass, the transmission shows higher volumetric power densities for lower gear ratios. A balanced ratio of the volumetric power density can be expected in the range of the overall gear ratio between 14 and 21.

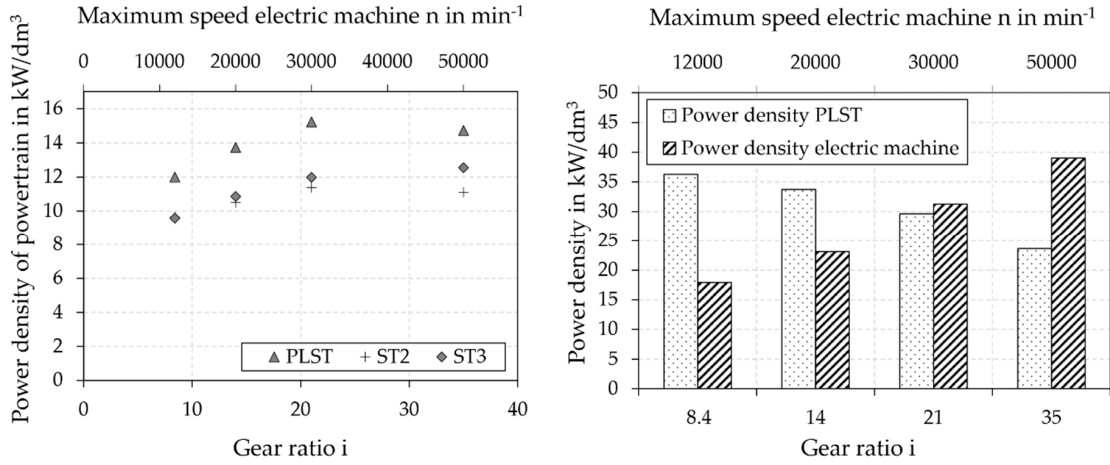

**Figure 22.** Volumetric power density of powertrain (**left**), electric machine and transmission (**right**).

## 6. Conclusions

This study investigated the influence of maximum speed of the electric machine on the power density of electromechanical powertrains. For this purpose, powertrains with an electric machine and suitable transmission were designed for different maximum speeds of the electric machines. An increase in the speed of the electric machine results in a higher required gear ratio of the transmission to reach the same transmission output conditions. Four different electric machines with maximum speeds from 12,000 $min^{-1}$ to 50,000 $min^{-1}$ were designed. This defined four powertrain designs, which are completed by suitable transmissions with three different configurations. The machines and

transmissions were calculated based on conceptual design. Further investigations and adaptions may be necessary to meet all requirements of a detailed design.

The results of the powertrain designs show that the mass of the electric machines decrease significantly with increasing maximum speed. Contrarily, the mass of the transmissions increases, whereby the increase of mass is less than the reduction in mass for the electric machines. As a result, the gravimetric power density can be increased significantly by increasing the maximum speed of the electric machines. Compared to the reference vehicle, which has a maximum speed of 11,400 $min^{-1}$, the gravimetric power density could be increased by 125.7% from 1.85 to 4.17 kW/kg by increasing the maximum speed up to 50,000 $min^{-1}$. In addition to the mass, the volume of the powertrains can also be reduced, whereby the reduction is smaller due to a disproportionately increase of the transmission volume.

For both parameters, the gravimetric and volumetric power density, a convergent growth could be found. In case of the volumetric power density, a peak can be determined between 20,000 $min^{-1}$ and 30,000 $min^{-1}$, while the peak of the gravimetric power density lies above the maximum speed of 50,000 $min^{-1}$ of the fastest electric machine considered in this study. The results of this study clearly show the potential of increasing both the gravimetric and volumetric power density of electromechanical powertrains significantly by increasing the maximum speed of the electric machine.

## 7. Outlook

In addition to the power density of electromechanical powertrains, there are other important parameters whose behavior for increasing maximum speed should be investigated in future studies.

The increasing speed limits the bore volume, due to mechanical stress in the rotor and bend critical speeds. Consequently, the output power of the machine is limited [5]. These limits have not been exceeded in the presented machines. Thus, the increase of the output power at high maximum speed will lead to a limitation in the power of the electric machine. Further investigation needs to be carried out to distinguish these limits. Furthermore, effects such as field weakening behavior in case of the electric machines or the manufacturing need to be discussed further on.

One crucial aspect in the design process of electromechanical powertrains is the efficiency behavior of the overall powertrain with increasing maximum speed that determines the size of the battery or the range of the car. In the electric machine, the hysteresis and eddy-current losses rise due to the higher frequency. Furthermore, the frequency dependent ohm's losses need to be investigated in depth to make a statement about the efficiency of the electric machine. For the transmission, the impact of the higher speeds on the losses has to be investigated. Concerning the no-load losses, higher speeds of the bearings and gearings are expected to have an impact towards higher losses. Optimizing the oil supply and oil flow as well as low viscosity fluids in the transmission can reduce the increase of these losses. In case of load-dependent losses, the increased speed causes reduced normal force on the tooth flanks due to the reduced torque and therefore reduced bearing loads, reducing the load-dependent losses. Also, high circumferential speeds can support fluid film lubrication regime of tribological contacts. Further investigations on the efficiency are to be carried out to describe the interdependency of higher circumferential speeds and the reduced forces acting in transmissions and powertrains with increasing maximum speed. In addition, the potential of low-loss gears and innovative base oils like water-containing gear fluids have to be evaluated, also in the context of holistic thermal managements.

Another aspect is the NVH-behavior of the transmission and the electric machine. Even though the eigenfrequencies of the electric machine are increasing, due to the more compact design, the frequency range of the exiting forces' waves increases as well with the maximum speed. For the transmission, it becomes more difficult to operate the first stages subcritically over the entire operating range. The interaction of the electric machine with its larger frequency range can also lead to a more complex excitation behaviour for the transmission and powertrain system, which should be examined in detail.

Furthermore, the increase of the maximum speed result in the use of new technologies, which currently may be not present or lead to increased costs of the powertrain system. For the electric

machine, one technology leap is the use of surface mounted magnets and the fixation with the bandage onto the rotor. The bandage material is quite expensive and the shrinkage on the rotor a challenging process. Furthermore, the amount of magnet material increases to keep the magnetic flux density in the air gap high with increasing size of the air gap. To achieve the high rotor speeds, bearings with higher accuracy and special materials to limit the centrifugal forces are needed. The lower torque leads to smaller sizes of the gearings, which means that influences from manufacturing errors are more important. Therefore, the quality of the gearings may be increased in order to obtain a sufficient load capacity and acoustical behaviour of the meshes under load.

**Author Contributions:** Conceptualization: D.S. and M.E.G.; methodology, D.S. and M.E.G.; software, D.S., A.T. and M.E.G.; validation, D.S., A.T., and M.E.G.; investigation, B.M., A.H.; resources, Institute of Machine Elements (TUM, Munich), Institute for Drive Systems and Power Electronics (LUH, Hannover); data curation, D.S. and M.E.G.; writing—original draft preparation, D.S., B.M., M.E.G. and A.H.; writing—review and editing, T.L., M.O., K.S. and B.P.; visualization, D.S. and M.E.G.; supervision, T.L., M.O., K.S. and B.P.; project administration, K.S.; All authors have read and agreed to the published version of the manuscript.

**Funding:** Supported by: Federal Ministry of Economic Affairs and Energy on thebasis of a decision by the German Bundestag.

**Conflicts of Interest:** The authors declare no conflict of interest.

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
