# Peer review of "On the Impact of Maximum Speed on the Power Density of Electromechanical Powertrains"

_vehicles, doi:10.3390/vehicles2020020_

Round 1

Reviewer 1 Report

 The paper formal structure is in very good condition.The presented results are very interesting and beneficital for this reserach branch.
My recommendation is to accept the manuscript for publishing.

Author Response

Thank you very much for the good feedback

English language and style:

We checked the publication once again for spell errors and style. We found few errors (dive -> drive, ...) and corrected it.

Reviewer 2 Report

Dear authors,

Interesting work! After reading through the paper, here are the required works to be done:

1) 2.1.1 Rotor Topologies

"Depending on the desired maximum surface speed, surface PM rotors are preferable to interior PM rotors."

Why does SPM is preferable to IPM? The benefit of using IPM for high speed applications is because the magnets are directly protected by the iron. It would be better for the authors to further justify the statement above with more technical explanation or references.

2) 2.1.6 Winding Technologies

"The random winding,....process is easier to automate."

Please provide the citations to support this statement.

3) Figure 3: Based on what the manuscript shows, this plot is from a paper cited in this work. Therefore, please put the citation number in the end of the figure caption.

4) Figure 9: Based on what the manuscript shows, this plot is from a paper cited in this work. Therefore, please put the citation number in the end of the figure caption.

5) Figure 16: The quality of this plot seems low. Please replace it with a higher quality one unless the quality of this plot in [53] is already this low.

Author Response

Reply to 1): Reference was added which supports the statement

Reply to 2): "easier to automate"  was changed to "fully automated" and reference  was added

Reply to 3): The figure is based on in house measurements of IAL institute

Reply to 4): Reference was added

Reply to 5): I tried to increase the quality of the plot. Unfortunately the quality in [55] is already pretty low. I removed the numbers on the axes, as they are not important for the paper. It's important to understand the way how the optimization for minimal mass of the transmission is done by adjusting the partial ratios. I think this picture is good to give an idea of the optimization process. If not, we can still remove the graph completely.

English language and style: Publication was again checked for spell errors and style. Some errors (dive->, ...) were corrected

Reviewer 3 Report

The subject of the article is very complex. It deals with the design of electric machines, as well as with bearings and lubrication, seals and power trains, and takes into account even travel comfort related to the noise. The complexity of the problem is reflected by the number of authors. Many aspects of the above problems were considered, however some were omitted, as e.g. bearing life related to increased speeds. The results obtained are very useful for further development of electric vehicles, which will undoubtedly continue in spite of its low perspective predicted by the critics of this trend of transport.

The English is good in spite of some minor mistakes, as e.g. "dive" systems in title of Tab. 3.

Author Response

English language and style: The publication was again checked for spelling errors and style. Some errors were found and corrected (e.g. dive->drive)

Round 2

Reviewer 2 Report

Dear authors,

Thank you for your work on revising the manuscript. My concerns are all addressed.